# Bone marrow adipose tissue is a unique adipose subtype with distinct roles in glucose homeostasis

Karla J. Suchacki [1], Adriana A. S. Tavares[1], Domenico Mattiucci[1,2], Erica L. Scheller [3],
Giorgos Papanastasiou [4], Calum Gray [4], Matthew C. Sinton [1], Lynne E. Ramage[1], Wendy A. McDougald[1],
Andrea Lovdel [1], Richard J. Sulston[1], Benjamin J. Thomas [1], Bonnie M. Nicholson[1], Amanda J. Drake [1],
Carlos J. Alcaide-Corral [1], Diana Said[1], Antonella Poloni [2], Saverio Cinti [2,5], Gavin J. Macpherson[6],
Marc R. Dweck[1], Jack P. M. Andrews [1], Michelle C. Williams [1], Robert J. Wallace [7],
Edwin J. R. van Beek [4], Ormond A. MacDougald[8], Nicholas M. Morton [1], Roland H. Stimson [1] &
William P. Cawthorn [1✉]

Bone marrow adipose tissue (BMAT) comprises >10% of total adipose mass, yet unlike white or brown adipose tissues (WAT or BAT) its metabolic functions remain unclear. Herein, we address this critical gap in knowledge. Our transcriptomic analyses revealed that BMAT is distinct from WAT and BAT, with altered glucose metabolism and decreased insulin responsiveness. We therefore tested these functions in mice and humans using positron emission tomography-computed tomography (PET/CT) with [18]F-fluorodeoxyglucose. This revealed that BMAT resists insulin- and cold-stimulated glucose uptake, while further in vivo studies showed that, compared to WAT, BMAT resists insulin-stimulated Akt phosphorylation. Thus, BMAT is functionally distinct from WAT and BAT. However, in humans basal glucose uptake in BMAT is greater than in axial bones or subcutaneous WAT and can be greater than that in skeletal muscle, underscoring the potential of BMAT to influence systemic glucose homeostasis. These PET/CT studies characterise BMAT function in vivo, establish new methods for BMAT analysis, and identify BMAT as a distinct, major adipose tissue subtype.

[1] University/BHF Centre for Cardiovascular Science, University of Edinburgh, The Queen's Medical Research Institute, Edinburgh BioQuarter, 47 Little France Crescent, Edinburgh EH16 4TJ, UK. [2] Dipartimento di Scienze Cliniche e Molecolari, Clinica di Ematologia, Università Politecnica delle Marche, Ancona, Italy. [3] Division of Bone and Mineral Diseases, Department of Medicine, Washington University, St. Louis, MO, USA. [4] Edinburgh Imaging, University of Edinburgh, Edinburgh, UK. [5] Dipartimento di Medicina Sperimentale e Clinica, Center of Obesity, Università Politecnica delle Marche, Ancona, Italy. [6] Department of Orthopaedic Surgery, Royal Infirmary of Edinburgh, Edinburgh, UK. [7] Department of Orthopaedics, The University of Edinburgh, Edinburgh, UK. [8] Department of Molecular & Integrative Physiology, University of Michigan, Ann Arbor, MI, USA. ✉email: W.Cawthorn@ed.ac.uk

Adipose tissue (AT) plays a fundamental role in systemic energy homoeostasis. In mammals it is typically classified into two major subtypes: white AT (WAT), which stores and releases energy and has diverse endocrine functions; and brown AT (BAT), which mediates adaptive thermogenesis[1]. Cold exposure and other stimuli also cause the emergence of brown-like adipocytes within WAT, typically referred to as beige adipocytes[1]. White, brown and beige adipocytes have attracted extensive research interest, owing largely to their roles and potential as therapeutic targets in metabolic diseases.

Adipocytes are also a major cell type within the bone marrow (BM), accounting for up to 70% of BM volume. Indeed, this BMAT can represent over 10% of total AT mass in healthy adults[2]. BMAT further accumulates in diverse physiological and clinical conditions, including aging, obesity, type 2 diabetes and osteoporosis, as well as therapeutic contexts such as radiotherapy or glucocorticoid treatment. Strikingly, BMAT also increases in states of caloric restriction[3]. These observations suggest that BMAT is distinct from WAT and BAT and might impact the pathogenesis of diverse diseases. However, unlike WAT and BAT, the role of BMAT in systemic energy homoeostasis remains poorly understood.

The metabolic importance of WAT is highlighted in situations of both WAT excess (obesity) and deficiency (lipodystrophy), each of which leads to systemic metabolic dysfunction[1]. This largely reflects the key role of WAT as an insulin target tissue. Adipocyte-specific ablation of the insulin receptor (IR) in mice causes insulin resistance, glucose intolerance and dyslipidaemia[4,5]. Similar effects result from adipocytic deletion of *Slc2a4* (Glut4), the insulin-sensitive glucose transporter[6]. Conversely, adipocyte-specific overexpression of *Slc2a4* reverses insulin resistance and diabetes in mice predisposed to diabetes[7]. Thus, insulin-stimulated glucose uptake is fundamental to WAT's role in systemic metabolic homoeostasis.

In contrast to WAT, the defining function of BAT is in mediating adaptive thermogenesis via uncoupled respiration. This is driven by mitochondria expressing uncoupling protein-1 (UCP1), which are abundant in brown adipocytes. Cold exposure is the classical stimulator of BAT activity: cold-induced glucose uptake is a hallmark of BAT activation and can be quantified in vivo using positron emission tomography–computed tomography (PET/CT) with $^{18}$F-fluorodeoxyglucose ($^{18}$F-FDG)[1,8]. Cold exposure or chronic sympathetic stimulation exert similar effects on beige adipocytes, and activation of brown or beige adipocytes can enhance energy expenditure[1]. Consequently, the past decade has seen extensive interest in activating BAT, or promoting beiging of WAT, to treat metabolic disease[1].

Compared to WAT and BAT, study of BMAT has been relatively limited. However, given its abundance and clinical potential[3], BMAT is now attracting increasing attention, with several studies beginning to investigate its metabolic properties. BM adipocytes (BMAds) have been proposed to exist in two broad subtypes: 'constitutive' BMAds (cBMAds) appear as contiguous groups of adipocytes that predominate at distal skeletal sites, whereas 'regulated' BMAds (rBMAds) occur interspersed with the haematopoietic BM in the proximal and axial skeleton[9]. Both subtypes are morphologically similar to white adipocytes, with large unilocular lipid droplets; however, their lipid content differs, with cBMAds having a greater proportion of unsaturated fatty acids than rBMAds or white adipocytes[3]. Like white adipocytes, BMAds also produce adipokines such as leptin and adiponectin[10] and can release free fatty acids in response to lipolytic stimuli, albeit to a lesser extent than WAT[11,12]. This lipolysis resistance is more pronounced for rBMAds, underscoring the functional differences in BMAd subtypes.

Despite these advances in understanding of BMAT lipid metabolism, its insulin responsiveness and role in systemic glucose homoeostasis is poorly understood. PET/CT studies have demonstrated glucose uptake into whole bones or BM of animal models and humans[13–16], but uptake specifically into BMAT has not previously been examined. Whether BMAT is BAT- or beige-like is also debated[3]. UCP-1 positive adipocytes have been noted in vertebral BM of a young mouse[13] and as an incidental finding in one clinical case study[17], but most studies find very low skeletal UCP-1 expression[18,19]. It has been suggested that BMAT is BAT-like, albeit based only on transcript expression from whole bones[18]. Notably, no studies have fully investigated if BMAT has properties of BAT or beige AT in vivo. Together, it remains unclear if BMAT performs metabolic functions similar to WAT, BAT or beige AT.

Herein, we used transcriptomic analysis and $^{18}$F-FDG PET/CT to address these fundamental gaps in knowledge and thereby determine if, in vivo, BMAT has metabolic functions of WAT or BAT. Our studies in animal models and humans demonstrate that BMAT is molecularly and functionally distinct from WAT, BAT and beige AT, identifying BMAT as a unique class of AT. We show that basal glucose uptake in BMAT is greater than in WAT and can be greater than that in skeletal muscle, and establish methods for BMAT characterisation by PET/CT. Together, this knowledge underscores the potential for BMAT to influence metabolic homoeostasis and sets a foundation for future research to reveal further roles of BMAT in normal physiology and disease.

## Results

**BMAT is molecularly distinct from WAT, BAT and beige AT**. The functional hallmarks of WAT, BAT and beige AT are reflected on a molecular level, with each class having distinct transcriptomic profiles and characteristic marker genes[20–22]. Thus, to test if BMAT has distinct metabolic functions, we first compared the transcriptomes of whole BMAT and WAT from two cohorts of rabbits. Rabbits were used because their skeletal and BMAT characteristics share similarities with those of humans. For example, unlike rodents, both rabbit and human bones have haversian canals[23] and display extensive BMAT formation that is concentrated around the central vasculature within the long bones[24,25]. Rabbits also have the practical benefit of allowing isolation of relatively pure pieces of whole BMAT, which is not possible in mice[25]. The latter is evident histologically (Supplementary Fig. 1A): whereas BMAT from the proximal tibia (pBMAT) contains some contaminating red marrow, this is absent in BMAT from the distal tibial (dBMAT), radius and ulna (ruBMAT), which are morphologically similar to iWAT and gWAT. Principle component analysis of both cohorts identified BMAT as a distinct depot compared to gonadal WAT (gWAT) and inguinal WAT (iWAT) (Fig. 1a). However, BMAT from either rabbit cohort was not uniformly enriched for markers of brown or beige adipocytes (Fig. 1b, Supplementary Fig. 1B): although *SLC27A2* was significantly higher in BMAT than WAT from both cohorts, and *PPARGC1A* in BMAT from cohort 1, several other brown and/or beige markers were more highly expressed in WAT, while most such markers were not differentially expressed between BMAT and WAT in either cohort (Fig. 1b, Supplementary Fig. 1B). Thus, the transcriptomic distinction with WAT is not a result of BMAT being more brown- or beige-like. Instead, gene set enrichment analysis (GSEA) highlighted the potential for BMAT to have altered glucose metabolism and decreased insulin responsiveness compared to WAT (Fig. 1c, d, Supplementary Fig. 1C).

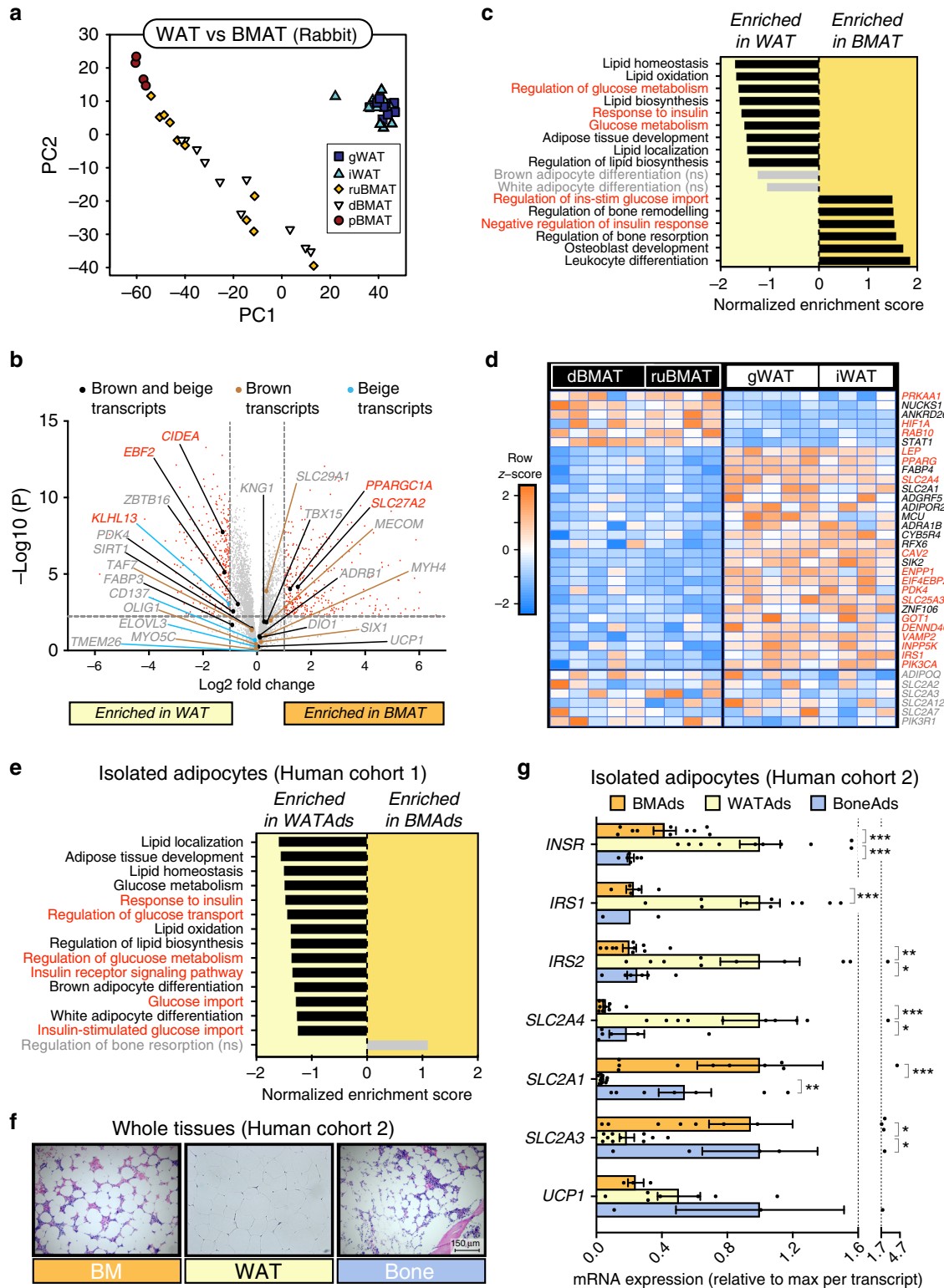

To determine if similar differences occur in humans, we next analysed the transcriptomes of adipocytes isolated from human femoral BMAT and subcutaneous WAT, which our previous analyses revealed to be globally distinct[26]. Consistent with our findings in rabbits, human BMAds were not enriched for brown or beige markers and had decreased expression of genes relating to glucose metabolism and insulin responsiveness (Fig. 1e, Supplementary Fig. 2A, B). To further address this we next

pursued targeted analysis of adipocytes isolated from a second cohort of subjects. White adipocytes were obtained from subcutaneous gluteofemoral WAT, while BMAds were isolated from the proximal femoral diaphysis; we also isolated adipocytes from trabecular bone of the proximal femoral metaphysis (Bone Ads) to assess potential site-specific differences in BMAd function[9]. Adipocyte purity was analysed histologically and by qPCR. As we have shown previously[26], histology confirmed the

**Fig. 1 BMAT is transcriptionally distinct from white, brown and beige adipose tissues. a–d** Transcriptional profiling of gonadal WAT, inguinal WAT and whole BMAT isolated from the proximal tibia (pBMAT), distal tibia (dBMAT) or radius and ulna (ruBMAT) of two cohorts of rabbits. **a** Principal component analysis of both cohorts. **b–d** Volcano plots (**b**), GSEA (**c**) and heatmaps (**d**) of transcripts differentially expressed between BMAT (dBMAT + ruBMAT) and WAT (iWAT + gWAT) in rabbit cohort 1. In **b–e**, red text indicates differentially expressed transcripts (**b**) or transcripts/pathways relating to glucose metabolism and/or insulin responsiveness (**c–e**); ns not significant. **e** Transcriptional profiling of adipocytes isolated from femoral BM or subcutaneous WAT of humans. **f** Representative micrographs of H&E-stained sections of human femoral BM, subcutaneous WAT and trabecular bone, representative of 24 subjects; scale bar = 150 μm. qPCR (**g**) of adipocytes isolated from tissues in (**f**). Data are mean ± SEM of the following numbers of human subjects per cell type: BM Ads, $n = 10$; WAT Ads, $n = 10$; Bone Ads, $n = 7$ (except *IRS1*, where $n = 2$ only). For each transcript, significant differences between each cell type are indicated by *$P < 0.05$, **$P < 0.01$ or ***$P < 0.001$. Significance for normally distributed transcripts (*INSR, IRS1, IRS2, SLC2A3* and *UCP1*) was assessed by one-way ANOVA using Dunnett's test for multiple comparisons; for *IRS1*, Sidak's multiple comparisons test was used because, with $n = 2$ for Bone Ads, only BMAds and WAT Ads were compared. For non-normally distributed transcripts (*SLC2A4* and *SLC2A1*) significance was assessed by the Kruskal–Wallis test, using Dunn's test for multiple comparisons. Source data are provided as a Source Data file. See also Supplementary Figs. 1 and 2.

presence of nucleated, lipid-laden adipocytes in the adipocyte fractions, while qPCR confirmed the expected separation of adipocytes from stromal-vascular cells, with haematopoietic and macrophage markers significantly enriched in the SVF of each depot (Supplementary Fig. 2C). In situ, these BM and bone adipocytes resembled unilocular white adipocytes (Fig. 1f); however, qPCR revealed significant differences in transcript expression of *INSR, IRS1, IRS2, SLC2A4, SLC2A1* and *SLC2A3* between WAT adipocytes and those from BM or bone (Fig. 1g). Notably, compared to white adipocytes, each BMAd subtype had decreased *SLC2A4* and increased *SLC2A1* and *SLC2A3*, suggesting that BMAds may have higher basal glucose uptake that is less insulin responsive. In contrast, there were no differences in expression of *UCP1*, and most other brown or beige adipocyte markers were not enriched in either BMAd subtype (Fig. 1g, Supplementary Fig. 2D).

Finally, we addressed if these differences also occur at the protein level. Antibodies against the IR, IRS1 and SLC2A4 did not work well for immunoblotting of rabbit samples; thus, we analysed expression of these proteins in gWAT, proximal tibial red marrow (pTib RM) and tibial dBMAT from rats. Adiponectin was readily detectable in dBMAT but not in RM, confirming purity of the isolated dBMAT (Fig. 2a, b). Notably, the IR precursor, IRS-1 and SLC2A4 were each expressed at significantly lower levels in dBMAT and RM than in gWAT (Fig. 2a, b). These observations therefore support the rabbit and human transcriptomic data by confirming that, compared to WAT, BMAT has lower expression of key insulin signalling components.

Taken together, these data demonstrate that BMAds in animal models and humans are molecularly distinct from white, brown and beige adipocytes, and suggest altered roles in systemic glucose homoeostasis and insulin responsiveness.

**BMAT in mice resists insulin-stimulated glucose uptake.** To test the metabolic functions of BMAT in vivo we first used [18]F-FDG PET/CT in mice to determine if, like WAT, BMAT is insulin-responsive. As expected, insulin decreased blood glucose (Fig. 3a) and increased [18]F-FDG uptake in the heart, iWAT, gWAT and skeletal muscle (Fig. 3b, c, e). Volumes of interest (VOIs) were drawn to assess [18]F-FDG uptake separately within bone and BM and thresholding was applied to separate bone from BM based on their different tissue densities. This revealed that insulin significantly increased [18]F-FDG uptake in femoral bone, whereas humoral bone uptake decreased; uptake in proximal or distal tibial bone was unaffected (Fig. 3e). To assess BMAT-specific [18]F-FDG uptake we took advantage of the regional differences in BMAT content around the mouse skeleton. Thus, adipocytes comprise only a small percentage of total BM volume of humeri, femurs and proximal tibiae, but predominate in distal tibiae (Fig. 3f). To address the contribution of BMAT to skeletal

[18]F-FDG uptake, we therefore quantified [18]F-FDG in a bone-region-specific manner to distinguish between areas of low BMAT (humerus, femur and proximal tibia) and high BMAT (distal tibia). This revealed that insulin did not significantly increase glucose uptake in any of the BM regions analysed (Fig. 3f). Surprisingly, insulin was associated with decreased glucose uptake in the humerus BM, which is BMAT-deficient. Thus, compared to WAT, BM and BMAT resist insulin-stimulated glucose uptake.

**BMAT in rats resists insulin-stimulated Akt phosphorylation.** We next investigated this insulin resistance at a molecular level. As shown in Figs. 1 and 2, expression of GLUT4 (*Slc2a4*) is significantly lower in BMAT than in WAT, which likely contributes to the impaired insulin-stimulated glucose uptake in BMAT. However, BMAT also has lower expression of the IR and IRS-1, suggesting further insulin resistance upstream of GLUT4. Insulin acts via the IR and IRS-1 to phosphorylate and thereby activate Akt, which is necessary for downstream induction of GLUT4 translocation to the plasma membrane[27]. Thus, we hypothesised that BMAT resists insulin-stimulated glucose uptake not only because of decreased GLUT4 expression, but also because of impaired Akt phosphorylation. To assess this in vivo, we treated rats with insulin or saline (vehicle control) and, 15 min post-treatment, isolated gWAT and tibial dBMAT for immunoblot analysis of Akt phosphorylation. As expected, in gWAT insulin significantly increased Akt phosphorylation at both S473 and T308 (Fig. 4a, b), confirming insulin responsiveness of WAT. In both control and insulin-treated rats, the absolute levels of Akt S473 and T308 phosphorylation were lower in dBMAT than in gWAT; however, in dBMAT stimulation of S473 phosphorylation was readily apparent (Fig. 4a, b). In contrast, in dBMAT insulin did not affect T308 phosphorylation, despite similar total Akt expression between these two tissues (Fig. 4a, b). These data confirm and extend our PET/CT results by demonstrating that BMAT resists insulin-stimulated phosphorylation of Akt at T308, a key step of the insulin signalling pathway.

**BMAT in mice is functionally distinct from BAT and beige AT.** To test if BMAT is BAT- or beige-like in vivo, we next analysed [18]F-FDG uptake following acute or chronic cold exposure in mice (Supplementary Fig. 3A). As shown in Supplementary Fig. 3B, acute (4 h) or chronic cold (72 h) increased energy expenditure (H4), reflecting the increased metabolic rate required to maintain body temperature. However, this occurred without causing weight loss or hypoglycaemia (Supplementary Fig. 3B–D), likely due to increased food consumption in chronic cold mice (Supplementary Fig. 3E). Acute or chronic cold exposure increased BAT [18]F-FDG uptake but did not increase uptake into skeletal muscle (Fig. 5a–c), indicating that non-shivering thermogenesis,

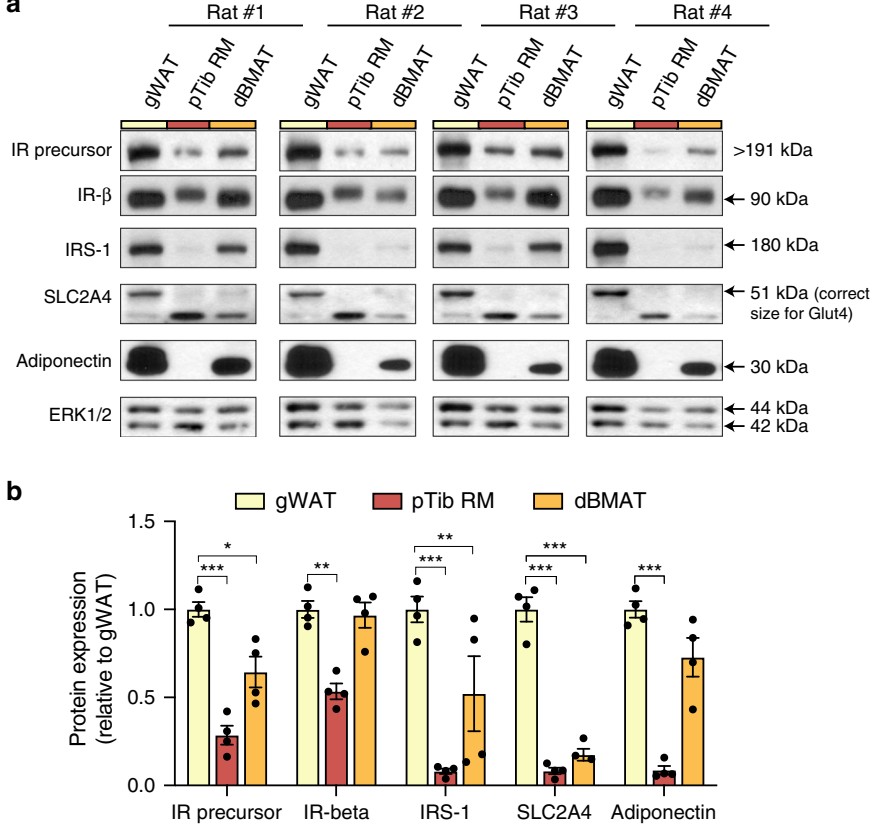

**Fig. 2 Expression of proteins essential for insulin signalling is lower in BMAT than in WAT.** Total protein was isolated from gWAT, proximal tibial RM (pTib RM) and tibial dBMAT of four rats, as described in the 'Methods'. **a** Expression of the indicated proteins was then determined by immunoblotting, with ERK1/2 used as a loading control. **b** Quantification of protein expression from (**a**). For each sample, expression of the indicated protein was normalised to ERK1/2 as a loading control. Data are presented relative to expression levels in gWAT as mean ± SEM of four rats. Significant differences between gWAT and pTib RM or dBMAT were determined by two-way ANOVA and are indicated by *$P < 0.05$, **$P < 0.01$ or ***$P < 0.001$. Source data and full scans of immunoblots are provided as a Source Data file.

rather than shivering, was likely responsible for the increased energy expenditure during cold exposure. Chronic cold also increased iWAT [18]F-FDG uptake, suggesting beiging of this depot (Fig. 5c). However, neither acute nor chronic cold exposure increased [18]F-FDG uptake into bone or BM (Fig. 5b). Indeed, cold exposure decreased [18]F-FDG uptake into distal tibial BMAT, highlighting fundamental differences with iWAT and BAT. Prolonged cold exposure can decrease tibial BMAT[28], which might explain the reduced [18]F-FDG uptake in tibiae of our cold-exposed mice. However, histomorphometry of proximal and distal tibiae revealed no differences in adipocyte density or BM adiposity between the three groups (Supplementary Fig. 4A–D). Cold exposure also decreased BAT lipid content and promoted beiging of iWAT, as indicated by formation of multilocular adipocytes (Fig. 5d, Supplementary Fig. 4E, F); however, these effects did not occur in proximal or distal tibial BMAT, in which no multilocular adipocytes were detectable (Fig. 5d). Consistent with this, in BAT and iWAT cold exposure induced brown and beige transcripts such as *Ucp1*, *Dio2*, *Prkaa1* and *Metrnl*[20,29], but this did not occur within bones (Fig. 5e–g, Supplementary Fig. 5A–C). Surprisingly, cold exposure did not upregulate the lipolysis-related transcripts *Pnpla2* or *Lipe* in BAT or iWAT (Supplementary Fig. 5A, B); however, this is consistent with several previous studies[30,31]. In contrast, within the bones cold exposure increased the expression of transcripts related to fatty acid oxidation (*Cpt1b* and *Ppara*), lipolysis (*Pnpla2* and *Lipe*) and lipogenesis (*Dgat2*) (Fig. 5g, Supplementary Fig. 5C), suggesting that

BMAds respond to cold exposure by increasing lipid turnover, even without increased glucose uptake or beiging.

Finally, one explanation for this lack of a beiging response in BMAT is that housing control mice at room temperature (RT) (22 °C) might have caused a mild cold stress, preventing detection of further beiging at 4 °C; however, even when compared to mice at thermoneutrality (28 °C), cold exposure did not induce a beiging response within bones (Supplementary Fig. 5D). Overall, these results show that, in vivo, BMAT is functionally distinct from brown and beige ATs.

**CT-based identification of BMAT in humans.** We next tested if these distinct metabolic properties extend to BMAT in humans. First, we established a method to identify BMAT from clinical PET/CT scans. To determine Hounsfield units (HU) for BMAT, we identified BMAT-rich and BMAT-deficient BM regions by magnetic resonance imaging (MRI). This revealed that sternal BM is BMAT-enriched while vertebral BM is BMAT-deficient (Fig. 6a); WAT was also analysed as an adipose-rich control region. We then co-registered the MRI data with paired CT scans of the same subjects (Fig. 6a). This revealed a distinct HU distribution for BMAT-rich sternal BM, intermediate between WAT and red marrow (RM) of BMAT-deficient vertebrae (Fig. 6b).

Using these distinct HU distributions, we generated a receiver operating characteristic (ROC) curve to identify optimal diagnostic HU thresholds to distinguish BMAT from RM

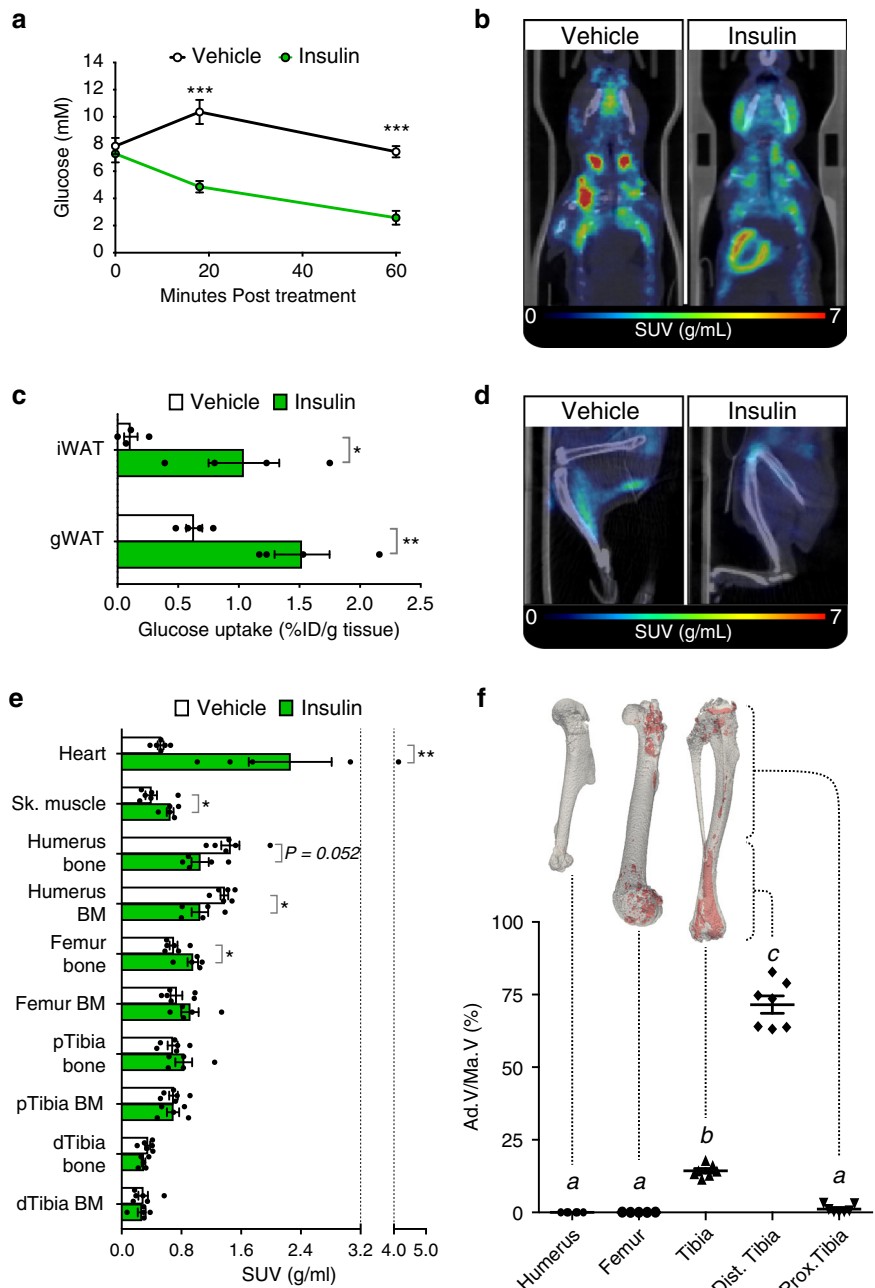

**Fig. 3 Insulin treatment in mice does not induce glucose uptake in BMAT.** Insulin-stimulated glucose uptake was assessed by PET/CT. **a** Blood glucose post-insulin (*n* = 7 mice) or vehicle (*n* = 8 mice). **b**, **d** Representative PET/CT images of the head and torso (**b**) or legs (**d**) of six vehicle- and five insulin-treated mice; some [18]F-FDG uptake into skeletal muscle is evident in the image of the vehicle-treated mouse (**d**), possibly resulting from physical activity. **c** Gamma counts of [18]F-FDG uptake in iWAT and gWAT of insulin- or vehicle-treated mice (*n* = 4 per group), shown as % injected dose per g tissue (% ID/g). **e** [18]F-FDG uptake in the indicated tissues of mice treated with vehicle (*n* = 6) or insulin (*n* = 5) was determined from PET/CT scans. **f** BMAT in humeri (*n* = 6), femurs (*n* = 5) or tibiae (*n* = 7) was analysed by osmium tetroxide staining. BMAT is shown in red in representative μCT reconstructions and quantified as adipose volume relative to total BM volume (Ad.V/Ma.V). Data in **a**, **c**, **e**, **f** are presented as mean ± SEM. Significant differences between control and insulin-treated samples are indicated by \**P* < 0.05, \*\**P* < 0.01 or \*\*\**P* < 0.001 and were assessed as follows: **a** repeated measures two-way ANOVA with Sidak's multiple comparisons test; **c** two-tailed unpaired *t* test with Holm–Sidak adjustment for multiple comparisons; **e** two-tailed Mann–Whitney test for tissues with non-normally distributed SUVs (heart, humerus bone, femur bone, femur BM and dTibia BM) and two-tailed unpaired *t* test for tissues with normally distributed SUVs (all other tissues). In **f**, significant differences were assessed by one-way ANOVA with Tukey's test for multiple comparisons. Groups in **f** do not significantly differ (*P* > 0.05) if they share the same letter. Source data are provided as a Source Data file.

(Fig. 6c). This revealed that BMAT-rich BM has HU < 115, whereas RM is mostly within 115–300 HU (Fig. 6b); bone was defined as >300 HU. To test the validity of these thresholds we applied them to clinical CT data to determine BMAT volume as % BM volume. We found that BMAT predominated in the arms,

legs and sternum but was markedly lower in the clavicle, ribs and vertebrae (Fig. 6d, e). Moreover, % BMAT showed age-associated increases in the axial skeleton but not in long bones (Fig. 6e). These data are consistent with previous studies showing that BMAT predominates in the long bones by early adulthood

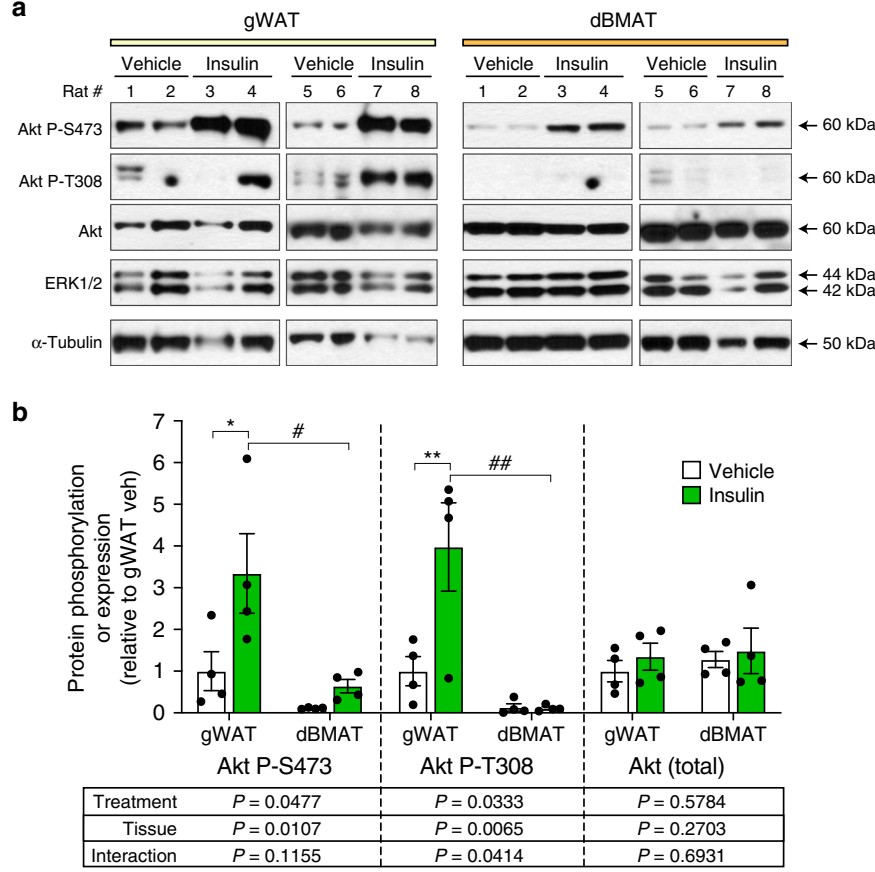

**Fig. 4 BMAT resists insulin-stimulated Akt T308 phosphorylation.** Male Sprague–Dawley rats at 13- to 15-weeks of age were fasted overnight prior to intra-peritoneal injection of saline ($n = 4$) or 0.75 U/kg insulin ($n = 4$). Tissues were isolated 15 min post-injection and total protein isolated as described in the "Methods". **a** Expression of the indicated proteins was determined by immunoblotting, with ERK1/2 and α-Tubulin used as loading controls. **b** Quantification of protein expression from (**a**). Expression of Akt P-S473 and Akt P-T308 was normalised to total Akt. Expression of total Akt was normalised to the average of ERK1/2 and α-Tubulin. Data are presented relative to gWAT of vehicle-treated rats as mean ± SEM. For each protein readout the influence of treatment or tissue, and interactions between these, was determined by two-way ANOVA, with $p$ values shown beneath the graph. Significant effects of treatment (within each tissue) or tissue (within each treatment) were further assessed by multiple comparisons and are indicated as follows: insulin vs vehicle, *$P < 0.05$, **$P < 0.01$; gWAT vs. dBMAT, #$P < 0.05$ or ##$P < 0.01$. Source data and full scans of immunoblots are provided as a Source Data file.

but continues to accumulate in axial bones beyond 60 years of age[32–34]. Together, this supports the validity of our CT thresholds for BMAT identification in humans.

**Human BMAT is functionally distinct from BAT.** We then applied these thresholds to human co-registered PET/CT data to assess [18]F-FDG uptake in BMAT, RM and bone. To test if BMAT is BAT-like we first compared BMAT [18]F-FDG uptake between three groups: subjects with no detectable supraclavicular BAT at RT (No BAT), subjects with active BAT at RT (Active BAT), and cold-exposed subjects (17 °C for 2 h; Cold). PET/CT confirmed BAT [18]F-FDG uptake in the latter two groups but not in the No BAT group (Fig. 7a, b, Supplementary Fig. 6A). The Cold subjects had slightly less humeral BMAT than subjects in the No BAT group, but BM adiposity at other sites did not significantly differ between the three groups (Fig. 7a, Supplementary Fig. 6B). Cold exposure did not alter [18]F-FDG uptake in scWAT but was associated with increased uptake in sternal and clavicular bone tissue; however, these were the only skeletal sites at which [18]F-FDG uptake significantly differed between the No BAT, Active BAT and Cold subjects (Fig. 7b–d). Indeed, the Active BAT and Cold subjects did not have increased glucose uptake in BMAT or

RM of any sites analysed (Fig. 7b–d). Thus, consistent with our findings in mice, BMAT glucose uptake in humans is not cold-responsive.

Our previous human PET/CT studies revealed that glucocorticoids acutely activate BAT[8]. Glucocorticoids also promote BMAT accumulation, demonstrating that BMAT can be glucocorticoid-responsive[3]. Therefore, to further test if BMAT shares properties of BAT, we analysed PET/CT data from previously reported subjects[8] to determine if glucocorticoids also influence glucose uptake in human BMAT. We found that prednisolone significantly influenced [18]F-FDG uptake only in vertebrae, in which there was a trend for increased uptake into RM but not BMAT or bone (Supplementary Fig. 6C). However, prednisolone did not influence [18]F-FDG uptake at any other site. Thus, unlike BAT, BMAT glucose uptake is not glucocorticoid-responsive.

**BMAT is a major site of basal glucose uptake in humans.** The above findings confirm that, in humans, BMAT is functionally distinct from BAT. However, while analysing these data, two other phenomena became apparent. Firstly, within axial bones of each subject, BMAT had significantly higher glucose uptake than bone (Fig. 7e); in the sternum, glucose uptake was also greater in

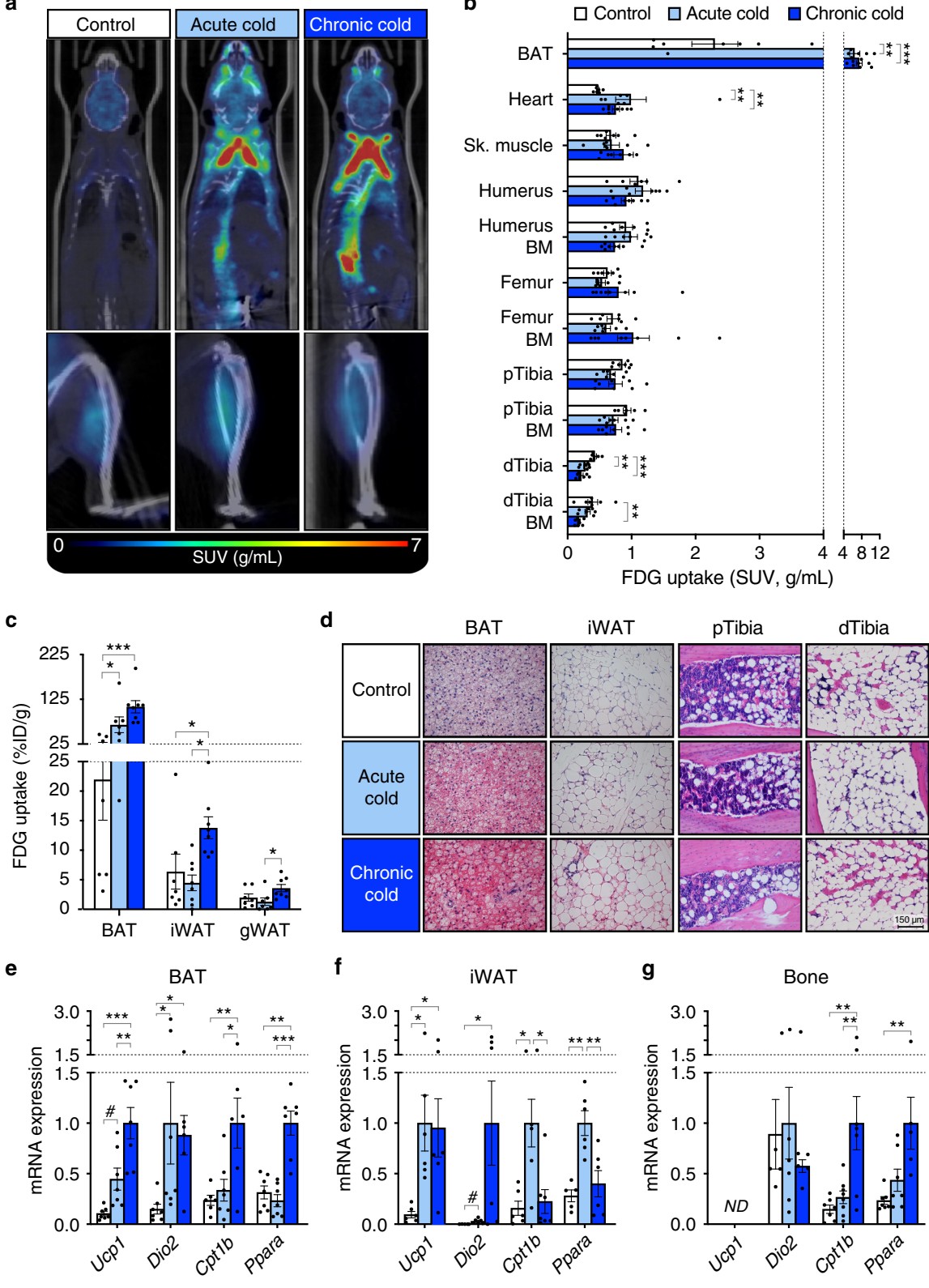

BMAT than in RM (Fig. 7e). Secondly, BMAT at each skeletal site had higher glucose uptake than scWAT, while uptake in axial BMAT was even greater than for skeletal muscle (Fig. 7f). Thus, despite being unresponsive to insulin or activators of BAT, BMAT has high basal glucose uptake, suggesting that it may have the potential to influence systemic glucose homoeostasis.

## Discussion

Unlike WAT and BAT, the role of BMAT in systemic energy metabolism is poorly understood. Previous studies have shed some light on BMAT lipid metabolism in vivo[11,12], and PET/CT has been used to assess glucose uptake into bones or BM[13–16]; however, our study is the first to characterise in vivo glucose

**Fig. 5 Cold exposure does not induce glucose uptake or beiging in BMAT.** Cold-induced glucose uptake was assessed by PET/CT, as described in Supplementary Fig. 3A. **a** PET/CT images representative of 7 control, 7 acute and 8 chronic cold mice show increased [18]F-FDG uptake in interscapular and paraspinal BAT but not in tibiae; some [18]F-FDG signal is evident in skeletal muscle of each group. **b, c** [18]F-FDG uptake in the indicated tissues was determined by PMOD analysis of PET/CT scans (**b**) or gamma counting (**c**). **d** Representative micrographs of H&E-stained tissues, showing that cold exposure decreases lipid content in BAT and promotes beiging of iWAT, but these effects do not occur in BMAT; scale bar = 150 μm. **e-g** Effects of cold exposure on expression of transcripts relating to brown and beige adipocyte function (*Ucp1* and *Dio2*) and fatty acid oxidation (*Cpt1b* and *Ppara*) in BAT, iWAT and whole bones. ND not detectable. Data in **b, c, e-g** are shown as mean ± SEM of the following numbers of mice per group: Control, n = 7 (**b, c, e, g**) or 6 (**f**); acute cold, n = 7 (**b, e**), 8 (**c, g**) or 6 (**f**); chronic cold, n = 8 (**b, c, g**) or 7 (**e, f**). Within each tissue, significant differences between groups are indicated by #P < 0.01, *P < 0.05, **P < 0.01 or ***P < 0.001. The following groups of data are non-normally distributed and were assessed using the Kruskal–Wallis test with Dunn's test for multiple comparisons: **b** Heart, Sk. Muscle and Femur bone; **c** iWAT and gWAT; **e** *Dio2*; **f** *Dio2* and *Cpt1b*; **g** *Dio2* and *Ppara*. Data for all other tissues (**b, c**) or transcripts (**e-g**) are normally distributed and were assessed using one-way ANOVA with Dunnet's or Tukey's tests for multiple comparisons. Source data are provided as a Source Data file. See also Supplementary Figs. 3–5.

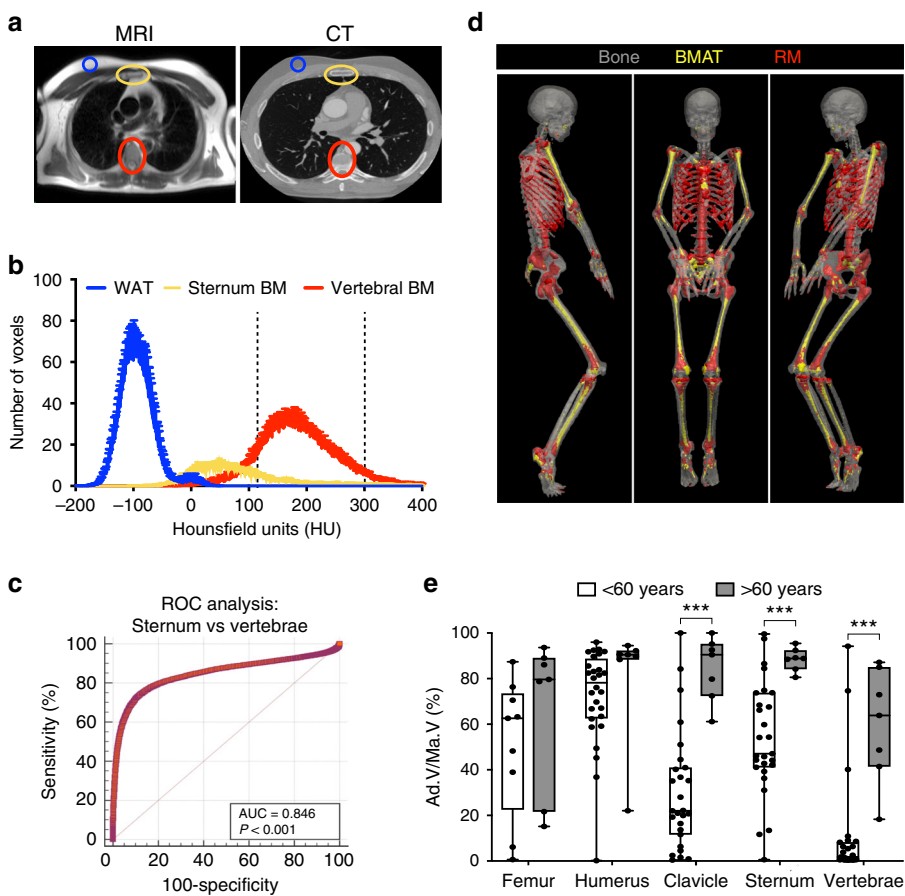

**Fig. 6 CT-based identification of BMAT in humans. a** Representative MRI (HASTE) and CT images from one subject. **b** HU distribution of scWAT, BMAT-rich BM (sternum) and BMAT-deficient BM (vertebrae). Data are mean ± SEM (n = 33). Thresholds diagnostic for BMAT (<115) and RM (115–300) are indicated by dashed lines. **c** ROC analysis to identify HU thresholds to distinguish BMAT-rich (sternum) from BMAT-deficient (vertebrae) regions of BM. **d** CT images of a 32-year-old subject, highlighting BMAT or RM identified using the diagnostic thresholds in (**b**); tibiae are shown for completeness but were not present in any other available CT scans. **e** Quantification of BMAT in CT scans of male and female subjects aged <60 or >60 years. A HU threshold of <115 was used to identify BMAT voxels in BM of the indicated bones, and total BM volume was also determined. The proportion of the BM cavity corresponding to BMAT (Ad.V/Ma.V) was then calculated. Data are shown as box-and-whisker plots of the following numbers of subjects for each group: <60 years, n = 28 (humerus), 9 (femur) or 27 (clavicle, sternum and vertebrae); >60 years, n = 7 for each bone; boxes indicate the 25th and 75th percentiles; whiskers display the range; and horizontal lines in each box represent the median. Significant differences between <60 and >60 groups are indicated by ***P < 0.001 and were assessed using a two-tailed Welch's *t* test for normally distributed data (Sternum) and a two-tailed Mann–Whitney test for non-normally distributed data (Femur, Humerus, Clavicle and Vertebrae). Source data are provided as a Source Data file.

metabolism specifically in BMAT. Our data provide key insights into how BMAT compares to WAT and BAT; reveal additional site-specific differences in BMAT characteristics; and identify BMAT as a major site of skeletal glucose disposal. Moreover, we establish a method for BMAT identification and analysis by PET/

CT that will open new avenues for future study of BMAT function.

We show that compared to WAT, BMAT resists insulin-stimulated glucose uptake. This is supported not only by PET/CT of mouse distal tibial BMAT, but also by the transcriptional

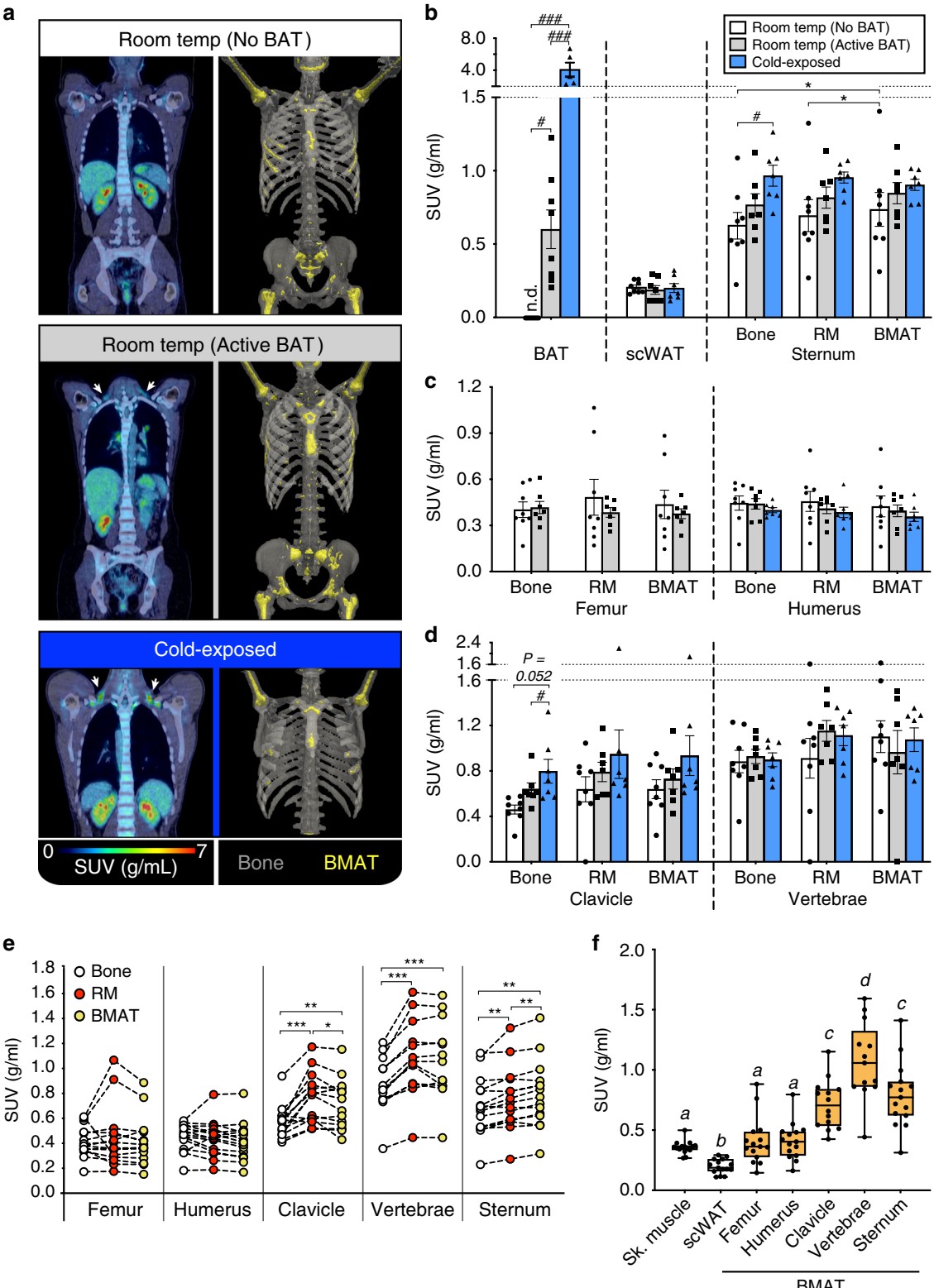

profiles of rabbit and human BMAT from other skeletal sites. This conclusion seemingly contrasts with findings elsewhere. For example, in humans, Huovinen et al. used PET/CT to assess BM [18]F-FDG uptake during hyperinsulinaemic euglycemic clamp, concluding that whole BM may be insulin-responsive[15]. Thus, one possibility is that insulin can stimulate BMAT glucose uptake under hyperinsulinaemic conditions. However, unlike our work, Huovinen et al. did not distinguish BMAT-rich from

BMAT-deficient BM, nor did they use a vehicle control to confirm if BM [18]F-FDG uptake is genuinely insulin-responsive. Indeed, microarrays show that *SLC2A4* and *IRS1* expression is negligible in human BM[35], while *Slc2a4* and *Irs1* are markedly lower in BM than in WAT or muscle of mice[36]. More-recent microarrays show that *Slc2a4, Insr, Irs1* and *Irs2* are lower in BMAds than epididymal white adipocytes of mice[37]. These data are strikingly consistent with our results for transcript expression in

**Fig. 7 Human BMAT is functionally distinct from BAT and is a major site of basal glucose uptake. a** Coronal PET/CT images (left side) and BMAT-thresholded CT images (right side) representative of 8 No BAT, 7 Active BAT and 7 Cold subjects. $^{18}$F-FDG uptake in BAT is evident in the Active BAT and Cold subjects (arrows). Femurs were not covered by the scans of the Cold group. **b–d** PET/CT analysis of $^{18}$F-FDG uptake in the indicated tissues of No BAT, Active BAT and cold-exposed subjects. **e, f** $^{18}$F-FDG uptake in bone tissue, RM and BMAT (**e**), or BMAT, scWAT and skeletal muscle (Sk. muscle) (**f**) of room-temperature subjects (No BAT and Active BAT groups). Data in **b–d** are mean ± SEM of 8 (No BAT) or 7 (Active BAT, Cold) subjects per group and were analysed by paired two-way ANOVA. Data in (**e–f**) are shown as paired individual values (**e**) or box-and-whisker plots (**f**) from 15 (Femur, Humerus, Sternum, Sk. Muscle and scWAT), 14 (Clavicle) or 13 (Vertebrae) subjects in the No BAT and Active BAT groups and were analysed by paired one-way ANOVA. In **f**, boxes indicate the 25th and 75th percentiles; whiskers display the range; and horizontal lines in each box represent the median. For **b–e**, significant differences between bone, RM and BMAT are indicated by *$P < 0.05$, **$P < 0.01$ or ***$P < 0.001$. Significant differences between No BAT, Active BAT and Cold groups are indicated by #$P < 0.05$ or ###$P < 0.001$. For **f**, tissues that do not share a common letter have significantly different SUVs. Source data are provided as a Source Data file. See also Supplementary Fig. 6.

rabbits and humans (Figs. 1, S1 and S2) and protein expression in rats (Figs. 2 and 4) and further support the conclusion that, compared to WAT, BMAT resists insulin-stimulated glucose uptake.

Our findings also have implications for BMAT lipid metabolism. Adipocyte-specific ablation of *Insr* in mice decreases BMAd size[4], suggesting a role for insulin in BMAd lipogenesis; however, it is unclear if this is through de novo lipogenesis from glucose, via insulin regulating uptake and esterification of fatty acids, or through stimulation by insulin of BMSC adipogenesis[38]. We show that BMAT is capable of insulin-stimulated Akt S473 phosphorylation, a modification implicated in lipogenesis downstream of mTORC2 activation[39]; however, it is unclear how this might stimulate lipogenesis in the absence of Akt T308 phosphorylation. One possibility is that, compared to WAT, BMAT lipogenesis is less dependent on insulin. Consistent with this, starvation decreases fatty acid esterification in WAT but not in BMAT[40]. This might explain why BMAT accumulates in hypoinsulinaemic conditions such as caloric restriction and in animal models of type 1 diabetes[3].

In addition to BMAT, we also find that insulin responsiveness varies among different bones: in insulin-treated mice, bone glucose uptake increases in femurs, decreases in humeri and is unaltered in tibiae. In contrast, Zoch et al report that insulin stimulates $^{18}$F-FDG uptake into whole femurs and tibiae[16]. This discrepancy may relate to technical differences: Zoch et al analysed whole bones (including BM) of anesthetised mice, whereas we distinguished between bone and BM and avoided anaesthesia. It is unclear why insulin is associated with decreased glucose uptake in humeral bone and BM; this is unlikely to be a technical issue given that we see the expected insulin-stimulated glucose uptake in the heart, skeletal muscle, WAT and femur. Thus, the lack of increases in humeri and tibiae suggests that there are site-specific differences in skeletal insulin responsiveness.

Another major finding is that BMAT is molecularly and functionally distinct from brown and beige ATs, both for cBMAT of mice and rabbits, and for rBMAT of humans at multiple skeletal sites. Functionally, we reveal that cold exposure does not stimulate BMAT glucose uptake in mice or humans, with mouse BMAT also showing no induction of the BAT and beige markers *Ucp1*, *Dio2*, *Prkaa1* and *Metrnl*. Indeed, we were unable to detect any multilocular BMAds in mice, regardless of the duration of cold exposure. In contrast, we found that chronic cold exposure in mice does increase expression of genes related to lipolysis, lipogenesis and fatty acid oxidation within the bones. This suggests that increased lipid catabolism may be balanced by increased lipogenesis, which might explain why 3-day cold exposure does not decrease BM adiposity. In contrast, Scheller et al.[28] showed that 21-day cold exposure decreases proximal tibial BMAT. Thus, longer durations of cold exposure may eventually exhaust compensatory increases in BMAT lipogenesis, resulting in decreased BM adiposity.

We further show that BMAT is distinct from BAT and beige AT at a molecular level, a finding coherent with several other studies. For example, we and others previously reported that tibial *Ucp1* expression is over 10,000-fold lower than in BAT[18,19], consistent with our present observation that *Ucp1* is undetectable in whole mouse bones. Similarly, microarrays show that *UCP1* is not enriched in whole BM of mice or humans[35,36], nor is it greater in BMAds vs white adipocytes of mice[37]. Consistent with this, recent work using lineage tracing and genetic models has convincingly demonstrated that BMAds do not express *Ucp1* during development or after adrenergic stimulation in mice[41]. Moreover, BMAd progenitors are more white-like than brown-like and do not express brown adipocyte markers after adipogenesis in vitro[42]. However, despite these diverse lines of evidence to the contrary, the concept that BMAT may be BAT- or beige-like has persisted. Thus, our in vivo functional analyses of mice and humans are a key advance because they confirm that cold exposure does not induce glucose uptake or beiging in BMAT. This demonstrates, conclusively, that BMAT is not BAT- or beige-like.

Our glucocorticoid studies provide further insights. Unlike in BAT, acute glucocorticoid treatment in humans does not stimulate glucose uptake in BMAT; however, it does influence uptake across lumbar vertebrae, with a trend for increases in RM (Supplementary Fig. 6C). It is notable that this occurs only in vertebrae, because these are also the bones in which glucocorticoids drive the greatest increases in fracture risk[43]. This raises the possibility that glucocorticoids modulate BM and bone metabolism in a site-specific manner and that these metabolic effects contribute to glucocorticoid-induced osteoporosis. Future studies using different doses and durations of glucocorticoids would further elucidate their ability to modulate metabolism of RM, BMAT and bone, and whether this influences glucocorticoid-induced osteoporosis.

Although BMAT glucose uptake is not stimulated by insulin at physiological concentrations, cold exposure or glucocorticoids, a major finding is that BMAT in humans has high basal glucose uptake, exceeding that of WAT and greater than that for skeletal muscle, bone or RM in the axial skeleton. Superficially, this seems at odds with two studies reporting that BM $^{18}$F-FDG uptake correlates inversely with BM fat content;[14,33] however, on further consideration, it is clear that these findings are not inconsistent with ours. Indeed, we show that axial bones have less BMAT but greater BM $^{18}$F-FDG uptake than humeri or femurs, mirroring these and other previous reports of $^{18}$F-FDG uptake in whole BM[15]. Importantly, unlike our approach, no previous studies have distinguished $^{18}$F-FDG uptake between RM and BMAT specifically. Thus, a unique advance of our work is the finding that, in axial bones, BMAT glucose uptake is greater than in bone and skeletal muscle and similar or greater than in RM (Fig. 7, Supplementary Fig. 6). This is supported by our mouse studies, which

show that relative glucose uptake in BM and BMAT is similar or greater to that in skeletal muscle (Fig. 3f, Fig. 5b). These observations confirm and extend previous studies showing that both BM and bone are sites of high glucose uptake, capable of exceeding levels observed in WAT or skeletal muscle[14–16]. Indeed, bone glucose uptake is required for normal metabolic function[44]. Together, these observations support the conclusion that BMAT may influence systemic glucose homoeostasis.

We also reveal that BMAT glucose uptake varies at different skeletal sites, generally being greater in axial BMAT compared to BMAT in long bones. This is consistent with depot-dependent differences in other BMAT characteristics and, broadly, with the concept that BMAT exists in regulated and constitutive subtypes[9]. However, while axial BMAT has higher glucose uptake, BMAT volume in peripheral bones is typically far higher (Fig. 7, Supplementary Fig. 6)[32]. Thus, the systemic metabolic influence of axial BMAT may be greater in conditions such as ageing, obesity, osteoporosis or caloric restriction, in which axial BMAT accumulates[3].

Related to this previous point, one remaining question is whether BMAT's high levels of relative glucose uptake (SUVs) translate into a similarly high capacity for total glucose uptake: based on the SUVs and the total volume of BMAT in humans, what is the total mass of glucose disposal within BMAT and how does this compare to other tissues? The ideal way to address this would be through dynamic total-body PET/CT scanning[45], but unfortunately this technology is still in its infancy and remains inaccessible to most researchers. Thus, to begin addressing this question we have estimated the total volume of BMAT, RM, bone and skeletal muscle in our human subjects, as described in the 'Methods'. By multiplying these values by the relevant SUV measurements (Fig. 7e, f), we estimate that BMAT has a total glucose uptake capacity that is up to 30% of that for skeletal muscle. This is greater than the uptake capacity of RM (5–10% of muscle uptake) but less than that of bone tissue (approximately 50–80% of muscle uptake). These estimates show that, in addition to having high SUVs for glucose, BMAT also has a relatively high absolute capacity for glucose uptake. However, it is important to emphasise that these are estimates; total-body PET/CT is needed to robustly calculate the glucose uptake capacity of BMAT and other tissues of interest, both in normal physiology and in clinical conditions. This will be another important goal for future studies.

Why does BMAT have such high basal glucose uptake? This may result from BMAds having high expression of *SLC2A1* and/or *SLC2A3* (Fig. 1), a finding supported by previous microarray studies[37]. Indeed, among numerous human tissues, *SLC2A3* expression is highest in BM[35], while *Slc2a3* is also greater in BM than in WAT of mice[36]. Mouse BMAds also have a dense mitochondrial network[46] and, by electron microscopy, we found that mitochondria are also abundant in human BMAds (S. Cinti, personal communication). This supports the conclusion that BMAds are metabolically active, which may further explain their high basal glucose uptake. Uptake and esterification of fatty acids is also greater in BMAT than in WAT;[40,47] however, the relative contribution of amino acids to BMAd energy requirements remains to be established. Future studies, including the use of respirometry, may prove extremely helpful in further dissecting these aspects of BMAd bioenergetics.

This high level of basal glucose uptake is also likely to be important for BMAd function. Ex vivo studies show that glucose incorporation into triglycerides is greater for BMAT than for gWAT[48], while whole BM displays very high rates of de novo lipogenesis from $^{14}$C-glucose in vivo[49]. This suggests that high basal glucose uptake can make a substantial contribution to lipogenesis in BMAT. Moreover, BMAd-derived fatty acids fuel the proliferation and survival of haematological tumours[50], and

fatty acid oxidation is important for maintenance of haematopoietic stem cells[51]. Based on these observations, we speculate that BMAds' high level of basal glucose uptake is required for de novo lipogenesis and the ability of BMAds to act as a local energy source to support haematopoiesis. The relative insulin resistance of BMAds may then allow them to maintain this essential function even in the face of systemic energy deficiency.

A final advance of our study is that we have developed a method to identify BMAT from CT scans, allowing its functional analysis by PET/CT. At least one other study has used HU thresholding to try to distinguish BMAT-enriched vs BMAT-deficient BM[52], but our method is more comprehensive because we directly compared paired MRI and CT scans to identify the optimal BMAT HU thresholds. The finding that the sternum is BMAT-rich was unexpected as this contrasts with most other axial bones; however, it is consistent with adipogenic progenitors being readily detectible within sternal BM[42]. Otherwise, our method identifies site- and age-dependent differences in RM and BMAT that are in full agreement with previous studies[32–34]. Applying our PET/CT approach to other clinical and preclinical studies, including retrospectively, therefore holds great promise to reveal further physiological and pathological roles of BMAT. Importantly, the diversity of PET tracers could extend such studies far beyond glucose metabolism, allowing many other functions of BMAT to be addressed.

In summary, this study is the first to dissect BMAT glucose metabolism in vivo and identifies BMAT as a distinct, major subtype of AT.

## Methods
A table of key resources is included in the Supplement (Supplementary Table 1).

**Human subjects**. All human studies (cohorts 1–6) were done in compliance with all relevant ethical regulations; following approval by the relevant local and/or national ethics committees; and in accordance with the Declaration of Helsinki, with all patients providing written informed consent prior to any study procedures. Further details are provided in the Reporting Summary.

Studies involving human subjects in cohort 1 (Fig. 1e, Supplementary Fig. 2A, B) were approved the local ethical committee (300/DG) of the Università Politecnica delle Marche-Azienda Ospedali Riuniti (Ancona, Italy), as described previously[26].

Studies in human cohorts 2, 3, 4 and 6 were reviewed and approved by the South East Scotland Research Ethics Committee (REC). The study in human cohort 5 was approved by the Caldicott Guardian for NHS Lothian. Studies in cohort 2, for collection and analysis of human tissues (Fig. 1f, g, Supplementary Fig. 2C, D), were assigned ethics number 10/S1102/39.

Subjects in cohort 3, who underwent paired MRI and CT scans (Fig. 6a–c), were part of the SALTIRE 2 trial (NCT02132026) or PRE18FFIR trial (NCT02278211), in which participants were consented for anonymised data to be used in ethically approved studies. Only baseline measurements not related to the trial outcomes, and not any of the pre-specified endpoints from the PRE18FFIR and SALTIRE 2 trials, are reported herein. These paired MRI-CT studies were assigned REC number 16/SS/0166 and were further approved by the United Kingdom (UK) Administration of Radiation Substances Advisory Committee.

PET/CT studies in human cohort 4 (cold-exposed subjects; Figs. 6e, 7a, b; Supplementary Fig. 6A, B), cohort 5 (no BAT and active BAT subjects; Figs. 6d, e, 7a–f; Supplementary Fig. 6A, B) and cohort 6 (placebo vs. prednisolone treatment; Supplementary Fig. 6C) were further reviewed and approved by the University of Edinburgh and NHS Lothian Academic and Clinical Central Office for Research and Development (ACCORD). Studies in cold-exposed subjects (cohort 4) were assigned REC number 13/SS/0242. Studies in Cohort 5 were assessed and approved by The Caldicott Guardian for NHS Lothian. Studies involving placebo or prednisolone treatment (cohort 6) were assigned REC number 11/SS/0074 and were not considered a CTIMP (clinical trial of an investigational medicinal product) because prednisolone was given to determine its physiological effect on BAT activation, as opposed to as a treatment or diagnostic test for a medical condition.

Subject characteristics are as described previously for subjects in cohort 1 (BMAd and WAT Ad isolation)[26] and in cohort 4 (cold-exposed subjects)[8,53]. Table 1 provides characteristics for all other human subjects in cohorts 2, 5 and 6. Subject numbers for cohort 4 and 6 were based on power calculations, whereas cohort 5 subjects were identified retrospectively.

**Table 1 Human subject characteristics. Age and BMI are mean ± SD. ND, not determined.**

| Study | Number of Subjects (Male/Female) | Age | BMI | Diabetic (%) | Osteoporotic (%) |
|---|---|---|---|---|---|
| Cohort 2: BMAd isolation | 10 (4/6) | 67.1 ± 5.9 | 31.6 ± 6.9 | 0 | 0% |
| Cohort 3: MRI and CT | 33 (24/9) | 65.7 ± 8.1 | 29.1 ± 4.8 | ND | ND |
| Cohort 5: Room temp (no BAT) | 10 (2/8) | 51.5 ± 19.6 | 20.7 ± 2.4 | 0 | ND |
| Cohort 5: Room temp (active BAT) | 10 (2/8) | 51.1 ± 16.0 | 21.0 ± 2.2 | 0 | ND |
| Cohort 6: Prednisolone vs. placebo | 6 (6/0) | 22.1 ± 2.8 | 22.0 ± 2.1 | 0 | 0 |

**Animals**. Studies in New Zealand White rabbits were done in compliance with all relevant ethical regulations and were approved by the University of Michigan Committee on the Use and Care of Animals, with daily care overseen by the Unit for Laboratory Animal Medicine. Rabbits were housed at 22 °C on a 12 h light/dark cycle with free access to water and food, with monitoring and tissue isolation done as described previously[25]. For cohort 1 (Fig. 1), male rabbits (3.14 ± 0.19 kg, mean ± SD) were fed a high-fibre diet (cat. No 5326, LabDiet), receiving 100 g/day (31.91 ± 0.19 g/kg body mass/day; mean ± SD) until 22 weeks of age. For cohort 2 (Supplementary Fig. 1), male rabbits were fed the same high-fibre diet ad libitum (68.26 ± 4.82 g/kg body mass/day; mean ± SD) until 13 weeks of age. Rabbits in each cohort were then sedated with an intramuscular injection of ketamine (40 mg/kg) and xylazine (5 mg/kg) before euthanising by injection of pentobarbital (65 mg/kg) via the marginal ear vein. Tissues were then isolated based on our previous protocol[25], as follows. First, marrow cavities of tibiae, radii and ulnae were exposed by cutting bones using a Dremel rotary tool with a #409 cutoff wheel (Robert Bosch Tool Corporation, Addison, IL, USA); a constant drip of sterile USP-grade water was used during cutting to prevent overheating. Tibiae were bisected longitudinally and BMAT was then removed using a stainless steel spatula. To isolate BMAT from radii or ulnae, epiphyses were removed by lateral incisions with the Dremel tool, thereby allowing access to the marrow cavity. BM was then extruded by tracing the perimeter of the marrow cavity with a 2-in.-long 21-gauge needle and subsequently scraping the BM out using a stainless steel spatula. Tissue samples were immediately snap-frozen in liquid nitrogen for subsequent RNA isolation and molecular analysis.

Studies in rats were done in compliance with all relevant ethical regulations and were approved by the University of Michigan Committee on the Use and Care of Animals, with daily care overseen by the Unit for Laboratory Animal Medicine. Rats were housed at 22 °C on a 12 h light/dark cycle with free access to water and food, as indicated. For BMAT protein expression studies, as reported previously[28], 1-year-old female high-capacity and low-capacity runner rats were obtained from the University of Michigan rat recycling programme. For BMAT signalling studies, as reported previously[11], male Sprague–Dawley rats at 13- to 15-weeks of age were fasted overnight prior to intra-peritoneal injection of saline (vehicle) or 0.75 U/kg insulin (Humalin R, Eli Lilly). At 14 min post-injection rats were sedated with isofluorane and decapitated one minute later (15 min post-injection) for tissue isolation and analysis.

Studies in C57BL/6JCrl mice were done in compliance with all relevant ethical regulations under project licences granted by the UK Home Office, and were approved by the University of Edinburgh Animal Welfare and Ethical Review Board. Male C57BL/6J mice were bred in-house and housed at 22–23 °C on a 12 h light/dark cycle with free access to water and food, as indicated.

**Human cell and tissue isolation**. For cohort 1, isolation of adipocytes from femoral head BM (BMAds) or subcutaneous WAT (WAT Ads) was done as described previously[26], as follows. BMAds were isolated from the femoral head of patients undergoing hip-replacement surgery, with each femoral head cut into four parts. Subcutaneous WAT was obtained from non-obese male and female patients undergoing abdominal surgery. After prompt washing in DMEM (Biological Industries, cat. no. L0064-500, Milano, Italy), any visible blood vessels were removed and the scWAT was minced into smaller pieces. Bone and scWAT were incubated in a solution containing 1 mg/ml of collagenase type I (Gibco, Invitrogen, Milano, Italy) and 1% human albumin (Albital, Kedrion, Lucca, Italy) at 37 °C for 90 min. After collagenase digestion, samples were filtered through a 200 μm nylon sieve to remove stromal elements. Cells were then washed four times with DMEM and centrifuged at 250 × g for 5 min. Collection of the floating layer after each centrifugation provided a pure fraction of floating adipocytes and a pellet containing stromal cells. After the last centrifugation, the purity of isolated cells was confirmed by immunofluorescence staining with Nile Red (Sigma-Aldrich, St. Louis, MO). The floating adipocytes were then used for RNA isolation.

For cohort 2, adipocytes from BM, trabecular bone and WAT were isolated from patients undergoing hip-replacement surgery: BMAds were obtained from the proximal femoral diaphysis; trabecular bone adipocytes were from the proximal femoral metaphysis; and WAT Ads were from gluteofemoral subcutaneous WAT. Immediately after surgical isolation, tissues were washed and stored in ice-cold Dulbecco's phosphate-buffered saline (DPBS, 14190250, Gibco) for transport to a sterile tissue culture hood. Therein, DPBS was decanted through a sterile 300 μm nylon filter to remove blood, lipid and small debris. The remaining washed tissue was then transferred to a sterile, pre-weighed petri dish (100 mm) and tissue mass

recorded. A solution of collagenase type I (Worthington Biochemicals) was made at 1 mg/mL in Krebs–Ringer HEPES (KRH) buffer (120 mM NaCl, 2 mM KCl, 1 mM KH$_2$PO$_4$, 0.6 mM MgSO$_4$, 1 mM CaCl$_2$·2H$_2$O, 82 mM HEPES, 5.5 mM D-Glucose, 1% bovine serum albumin (BSA)) pre-warmed to 37 °C; sufficient volume was made to allow for 2 mL per mg tissue and the solution was passed through a 0.22 μm filter before use. After weighing, each tissue was minced in the petri dish using a sterile scalpel and scissors, then transferred to a Falcon tube containing the collagenase solution. Tissues in collagenase were then incubated for 45 min in a shaking water bath (120 rpm) at 37 °C. Next, collagenase-digested tissue was passed through a 300 μm nylon filter and the cells within the filtrate were washed with fresh KRH buffer. Samples were then centrifuged at 500×g for 5 min at 4 °C. The floating adipocyte layer was transferred by pipette to a new tube to be used for RNA isolation; an aliquot was also analysed histologically to confirm the presence of adipocytes. After aspirating and discarding the supernatant, the stromal vascular fraction (SVF) of cells within the pellet was resuspended in 2× volume of red blood cell lysis buffer (Cat. No. R7757, Sigma) and incubated at RT for 5 min to lyse erythrocytes. KRH buffer was added to bring the volume to 15 mL and samples were centrifuged at 700 rcf for 10 min at 4 °C. The SVF pellet was then used for RNA isolation.

**RNA isolation and reverse transcription**. For human cohort 1, RNA was extracted using an RNeasy mini kit (Qiagen, Milan, Itlay) according to the manufacturer's protocol. For human cohort 2 and mouse studies, RNA was isolated from cells or tissues using Ribozol™ solution (cat. No. N580, Amresco, USA,) according to the manufacturer's protocol. For rabbit studies, RNA was isolated from tissues using RNA STAT60 reagent (Tel-Test, Inc.) according to the manufacturer's protocol. Tissues included iWAT, gWAT, dBMAT and ruBMAT of both cohorts, and pBMAT of rabbit cohort 2. RNA was quantified using a NanoDrop spectrophotometer (Thermo Scientific, USA), and cDNA was synthesised using the Taqman® High Capacity cDNA Reverse Transcriptase Kit (Applied Biosystems, USA, cat. no. N8080234), in accordance with the manufacturer's guidelines.

**Microarray analyses**. For human cohort 1, RNA extraction, generation of single-strand biotinylated cDNA, and hybridisation to Human GeneChip® HTA 2.0 Arrays (Affymetrix) has been described previously[26] as follows. To obtain single-strand biotinylated cDNA, 100 ng of total RNA was subjected to two cycles of cDNA synthesis with the Ambion® WT PLUS Expression Kit (Affymetrix, Santa Clara, CA). First-cycle, first-strand synthesis involved using an engineered set of random primers that exclude rRNA-matching sequences and include T7 promoter sequences. cDNA resulting from second-strand synthesis was transcribed in vitro with T7 RNA polymerase to generate cRNA, which was subjected to second-cycle, first-strand synthesis in the presence of dUTP at a fixed ratio relative to dTTP. Single-strand cDNA was then purified and fragmented with a mixture of uracil DNA glycosylase and apurinic/apirimidinic endonuclease 1 (Affymetrix) in correspondence with incorporated dUTP. DNA fragments were terminally labelled with biotin by terminal deoxynucleotidyl transferase (Affymetrix). The biotinylated DNA was then hybridised to Human GeneChip® HTA 2.0 Arrays (Affymetrix), containing more than 285,000 full-length transcripts covering 44,700 coding genes and 22,800 non-coding genes selected from the Homo sapiens genome databases RefSeq, ENSEMBL, and GenBank. Chips were washed and scanned on the Affymetrix Complete GeneChip® Instrument System, generating digitised image data (DAT) files. The DAT files were analysed with the Affymetrix Expression Console software. The full dataset was normalised using the Robust Multi-alignment Algorithm (RMA). The expression values thus obtained were tested with R software (v2.15.0). Further normalisation steps included per-chip normalisation to the 50th percentile and per-gene normalisation to the median. Transcripts were considered to be significantly differentially expressed between BMAds and WAT Ads when they had an adjusted P value of 0.05 or less; P values were adjusted for multiple comparisons using the Benjamini and Hochberg approach to control for false-discovery rate[54].

For rabbit studies, purified RNA was digested on-column with DNase I and cleaned using the Qiagen RNeasy kit (Qiagen, Valencia, CA, USA) as recommended by the manufacturer. Total RNA was then submitted to the microarray core at the University of Michigan. The samples were screened for quality and processed in the microarray facility using custom rabbit Affymetrix arrays and the IVT Express kit (Affymetrix, Santa Clara, CA, USA). As a QC measure, the distribution of probe intensities and the 5′ to 3′ degradation profiles

were checked to be consistent across samples. The core's statistician used RMA, from the Affy package of Bioconductor (v2.11), to fit log$_2$ expression values to the data[55]. Weighted, paired, linear models were then fit and contrast computed using the limma package[56]. Weighting was done using a gene-by-gene algorithm designed to down-weight chips that were deemed less reproducible[57]. Probe-sets with a variance over all samples less than 0.05 were filtered out. Of the remainder, probe-sets with a log$_2$-fold change of 2 or greater and an adjusted $P$ value of 0.05 or less were retained. $P$ values were adjusted for multiple comparisons using the Benjamini and Hochberg false discovery rate approach[54]. Affy, affyPLM, and limma packages of Bioconductor, implemented in the R-statistical environment (v2.15.0) were used to analyse the data, including PCA analysis[55].

Pathways enriched in BMAT (combined dBMAT and ruBMAT) or WAT (combined iWAT and gWAT) of rabbits, or in isolated adipocytes from femoral BM and gluteofemoral scWAT (humans), were identified using GSEA software and the Molecular Signature Database (MSigDB)[58]. For rabbits, pBMAT was not included in these analyses because it contained a high proportion of RM and therefore represented a less pure BMAT sample[25]. To ensure maximum compatibility with this software, rabbit gene identifiers were first converted to their corresponding human homologues using the BetterBunny algorithm[59]. Volcano plots and heat maps (Pearson Distance) to visualise significantly differentially expressed transcripts (adjusted $P$ value < 0.05, fold-change > 2) were generated using Prism 8 (GraphPad) and Heatmapper software[60], respectively.

**qPCR.** For human cohort 2 and tissues from mice, reverse transcription, primer design/validation and qPCR were done as follows. RNA was extracted from tissue using RNA STAT60 reagent (Tel-Test, Inc.) or RiboZol reagent (AMRESCO LLC) according to the manufacturer's instructions. Synthesis of cDNA was done using TaqMan reverse transcription reagents (Thermo Fisher Scientific) using 1 µg of RNA template per reaction, as per manufacturer's instructions. Transcript expression was then analysed by qPCR in 10 µL duplicate reactions using 1–4 µL of cDNA template. Sybr Green-based qPCR was done using qPCRBIO SyGreen Blue Mix (part number PB20.17-20; PCR Biosystems, UK). Taqman-based qPCR was done using qPCRBIO Probe Blue Mix (part number PB20.27-20, PCR Biosystems, UK). Reactions were loaded into 384-well qPCR plates (part number 72.1985.202; Sarstedt, UK) and run on a Light Cycler 480 (Roche). Transcript expression was calculated based on a cDNA titration loaded on each plate. Expression of each target gene was normalised to expression of 18S rRNA (human gene, *RNA18SN5*; mouse gene, *Rn18s*), *IPO8* or *Ppia*, based on consistency of housekeeper expression across all samples. For each transcript, expression is presented relative to the group with the highest expression. Taqman assays (Thermo Fisher) were used to analyse mouse tissues for expression of *Ucp1* (cat. no. Mm01244861_m1), *Pnpla2* (cat. no. Mm00503040_m1), *Lipe* (cat. no. Mm00495359_m1) and *Dgat2* (cat. no. Mm00499536_m1), and expression of *ITGAM* (cat. no. Hs00167304_m1) and *PTPRC* (cat. no. Hs04189704_m1) in SVF and Ads from humans. All other primer sequences are described in Table 2.

**Protein isolation and western blotting.** To isolate proximal tibial RM and tibial dBMAT, both tibiae from each rat were cleaned of muscle and tendon using gauze and then cut axially at the tibia/fibula junction using a Dremel rotary tool with a Dremel 545 Diamond cutting wheel[28]. The marrow was removed from the proximal and distal tibial segments by centrifuging at 3000×$g$ for 1 min at 4 °C. The marrow plugs from the distal tibia was bisected horizontally and the most distal, 'white' portions pooled (two per animal) and used for protein extraction (dBMAT). Where indicated, the RM pellet from the proximal tibia and gWAT were also collected for analysis. Tissue was lysed at 4 °C on ice in SDS lysis buffer (1% sodium dodecyl sulfate (SDS), 0.1 M Tris pH 6.8, 13 mM ETDA, 1 mM PMSF, 1 mM sodium orthovanadate, 1 mM sodium fluoride) with protease inhibitor cocktail (Roche Cat:11836153001, 1 tablet/7 mL lysis buffer). WAT was homogenised using a Bullet Blender Gold (Next Advance Inc., Averill Park, NY, USA) while RM and BMAT were homogenised by passing through a series of sequentially smaller needles. After tissue disruption, the lysate was boiled at 95 °C on a dry heat block for 5 min. Boiled lysate was centrifuged at 16,000×$g$ and the cleared liquid fraction below the lipid layer and above the pellet was removed to a new tube for downstream analyses. Protein concentration was quantified using the BCA protein assay (Pierce 23225).

For SDS polyacrylamide gel electrophoresis, lysates were diluted to equal protein concentration in lysis buffer plus 1× NuPage LDS buffer (Invitrogen, Carlsbad, CA, USA) supplemented with 2.5% 2-mercaptoethanol. Samples were boiled for 5 min, cooled on ice for 1 min, vortexed, and equal amounts of protein (20 µg per lane) separated on gradient polyacrylamide gels (Invitrogen, Carlsbad, CA, USA). Samples were then transferred to Immobilon PVDF membranes (Millipore, Billerica, MA, USA). After transfer, equal protein loading between lanes was confirmed by Ponceau staining of membranes, after which membranes were blocked in 5% milk for 1 h at RT. Membranes were incubated in primary antibodies (Table 3; each in 5% BSA) overnight at 4 °C, followed by incubation with HRP-conjugated secondary antibodies (GE HealthCare, Waukesha, WI, USA) at 1:3,000 (anti-Rat) or 1:5,000 (anti-Rabbit) dilution for 1 h at RT. Secondary antibodies were then visualised with Western Lightning Plus (Perkin Elmer, Waltham, Massachusetts) or SuperSignal™ West Femto Maximum Sensitivity Substrate (Pierce, Rockford, IL, USA) titrated 1:1–1:3 with 1 M Tris-HCl pH 8.0 to achieve optimal signal. Band density was quantified in FIJI using the gel analyser plugin[61].

**Histology and histomorphometry.** Fixed murine and human soft tissue and decalcified bones (14% EDTA for 14 days) were paraffin embedded by the histology core at The University of Edinburgh's Shared University Research Facilities (SuRF). Paraffin-embedded tissue sections were then sectioned at 100 µm intervals using a Leica RM2125 RTS microtome and collected onto 76 ×26 mm StarFrost slides (VWR, UK). The slides were baked at 37 °C overnight before Hematoxylin and Eosin (H&E) staining.

Quantification of multilocular adipocyte frequency was performed on H&E-stained iWAT and BMAT from male C57BL/6J mice aged ~18 weeks. Adipocytes

**Table 2 Sequences of primers used for qPCR.**

| Species | Transcript | Forward Primer (5′-3′) | Reverse Primer (5′-3′) |
|---|---|---|---|
| *H. sapiens* | *ADIPOQ* | TCCTCACTTCCATTCTGACTGC | GTAGAACAGCTCCCAGCAACA |
| *H. sapiens* | *CPT1B* | CTGGTGCTCAAGTCATGGTG | CTGCCTGCACGTCTGTATTC |
| *H. sapiens* | *INSR* | TAGACGTCCCGTCAAATATTGC | GAAGAAGCGTAAAGCGGTCC |
| *H. sapiens* | *IPO8* | TTTCCCCTCAAATGTGGCAGC | CTTCTCCTGCATCTCCACATAGT |
| *H. sapiens* | *IRS1* | AGAGGACCGTCAGTAGCTCA | TCTCTCATGACACGGTGGTG |
| *H. sapiens* | *IRS2* | CACCTACGCCAGCATTGACT | GAAACAGTGCTGAGCGTCTTC |
| *H. sapiens* | *METRNL* | CCACAGGCTTCCAGTACGAG | TCAGGCTCGTGGGTAACTTG |
| *H. sapiens* | *PPARG* | TCATGCTTGTGAAGGATGCAAG | ATCCCCACTGCAAGGCATTT |
| *H. sapiens* | *PRDM16* | GAGGAGAGAGATTCCGCGAG | CCCGGTTGGGCTCATACAT |
| *H. sapiens* | *RNA18SN5* | CGATGCTCTTAGCTGAGTGT | GGTCCAAGAATTTCACCTCT |
| *H. sapiens* | *SLC2A1* | TCCCTGCAGTTTGGCTACAA | CAGGATGCTCTCCCCATAGC |
| *H. sapiens* | *SLC2A3* | TAGATTACAGCGATGGGGACAC | GTAGCCAAATTGGAAAGAGCCG |
| *H. sapiens* | *SLC2A4* | TCGGGCTTCCAACAGATAGG | GTTGTACCCAAACTGCAGGG |
| *H. sapiens* | *TGM2* | GGAGTATGTCCTCACCCAGC | CGTTCTTCAGGAACTTGGGGT |
| *H. sapiens* | *UCP1* | GTGTGCCCAACTGTGCAATG | ACGTTCCAGGATCCAAGTCG |
| *M. musculus* | *Cpt1b* | TGTCTACCTCCGAAGCAGGA | CGGCTTGATCTCTTCACGGT |
| *M. musculus* | *Dio2* | TCTTCCTGGCGCTCTATGAC | ACCACACTGGAATTGGGAGC |
| *M. musculus* | *Metrnl* | CCTGGAGCAGGGAGGCTTAT | TCGGACAACAAAGTCACTGGT |
| *M. musculus* | *Ppara* | CCTGAACATCGAGTGTCGAATAT | TCTTCTTCTGAATCTTGCAGCT |
| *M. musculus* | *Pparg* | GGAAAGACAACGGACAAATCAC | TACGGATCGAAACTGGCAC |
| *M. musculus* | *Ppia* | CACCGTGTTCTTCGACATCA | CAGTGCTCAGAGCTCGAAAGT |
| *M. musculus* | *Prkaa1* | ACCAGGTCATCAGTACACCATC | ACACCGGAAAGGATCTGCTG |
| *M. musculus* | *Rn18s* | CGATGCTCTTAGCTGAGTGT | GGTCCAAGAATTTCACCTCT |
| *M. musculus* | *Slc2a4* | ACTCATTCTTGGACGGTTCCTC | CACCCCGAAGATGAGTGGG |

**Table 3 Antibodies used for Western Blotting.**

| Protein | Antibody | Dilution | Band Size (kDa) |
|---|---|---|---|
| Insulin receptor-β | Santa Cruz, sc-711 | 1:500 | >191 (IR precursor); 90 (IR-beta) |
| IRS-1 | MilliporeSigma, 06-248 | 1:1000 | 180 |
| Glut4 | Cell Signalling, 2213 | 1:1000 | 51 |
| Adiponectin | Sigma, A6354 | 1:1000 | 30 |
| ERK1/2 | Cell Signalling, 9102 | 1:1000 | 42, 44 (double band) |
| P-Akt S473 | Cell Signalling, 9271 | 1:1000 | 60 |
| P-Akt T308 | Santa Cruz, sc-16646-R | 1:1000 | 60 |
| Akt | Cell Signalling, 4691 | 1:1000 | 60 |
| α-Tubulin | Invitrogen, MA1-80017 | 1:1000 | 55 |

were considered multilocular if they contained three or more smaller lipid droplets associated with a large lipid droplet, as previously described[11]. Both the total marrow area and the number and area of adipocytes were measured and expressed as the number of Ads per marrow area (N.Ad/Ma.Ar; number/mm²) and adipocyte area as a percentage of marrow area (Ad.Ar/Ma.Ar; %) using image J[61].

**Mouse insulin-treatment studies.** C57BL/6J male mice aged ~16 weeks were fasted for 4 h at RT. Insulin (Humulin S, Eli Lilly; 0.75 mIU/g body mass) or sterile saline (0.9%) was then adminstred to to mice via intraperitoneal injection immediately prior to ¹⁸F-FDG injection. Mice were then returned to their cages. At 0 min (just before ¹⁸F-FDG injection), 15- and 60-min post-¹⁸F-FDG blood glucose was measured by tail venesection and blood sampled directly into EDTA-microtubes (Sarstedt, Leicester, UK). Mice were then anaesthetised and ¹⁸F-FDG distribution assessed by PET/CT. After PET/CT, mice were sacrificed by overdose of anaesthetic. BAT, iWAT, gWAT, pWAT, mWAT, gonads, brain, kidneys, liver, spleen, pancreata, heart, soleus, gastrocnemius, femur, tibiae, humeri and tail vertebrae were then dissected and ¹⁸F-FDG uptake into each tissue was determined using a gamma counter (PerkinElmer). Counts per minute were converted to MBq activity using a standard conversion factor calibrated for the gamma counter. MBq were then corrected for radioactive decay based on the time of ¹⁸F -FDG administration and the time of gamma counting. Finally, the corrected MBq values were normalised to the mass of each tissue. The final gamma counts are therefore presented as % injected dose per g tissue (%ID/g). Frozen and fixed tissues were analysed separately and the average MBq per tissue was then calculated. Half of the dissected material was then snap frozen on dry ice and stored at −80 °C for molecular analyses. The remaining half of the dissected material was placed into 10% formalin and stored at 4 °C for histological analysis. PET/CT analysis was then done as described below.

**Mouse cold-exposure studies.** The protocol is adapted from ref. [62], with a summary depicted in Supplementary Fig. 3A. For the acute and chronic cold exposure studies (Fig. 5, Supplementary Figs. 3–5), male C57BL/6J mice aged ~18 weeks were housed individually in TSE PhenoMaster cages for indirect calorimetry, monitoring of physical activity, and measurement of *ad libitum* food and water consumption. Mice in each group were first housed in these cages for 3 days at RT for acclimation and baseline measurements. Group 1 (Control) were then housed for 72 h at RT in standard cages; group 2 (Acute cold) for 68 h at RT in standard cages, followed by 4 h at 4 °C in TSE cages; and group 3 (Chronic cold) for 72 h at 4 °C in TSE cages. Following TSE housing at 4 °C, Acute cold and Chronic cold mice were returned to standard cages that had been pre-cooled on ice to 4 °C; Control mice continued to be housed in standard cages at RT. All groups were fasted, with access to water, for 4 h before administration of ¹⁸F -FDG (such that Acute cold mice were fasted throughout their 4 h cold exposure). Cages of cold-exposed mice were stored on ice in a ventilated cooler for transport to the PET/CT facility, while Control mice were transported at RT. After intraperitoneal injection of ¹⁸F -FDG, mice were returned to cages at RT (Control) or 4 °C (Acute and Chronic cold). At 0 min (just before ¹⁸F-FDG injection), 15- and 60-min post-¹⁸F -FDG, blood glucose was measured by tail venesection. At 60-min post-¹⁸F-FDG, mice were placed under general anaesthesia and underwent PET/CT imaging. Euthanasia, tissue isolation and gamma counting were done as described above for the insulin-treatment studies. PET/CT analysis was then done as described below.

To assess effects of cold exposure compared to mice housed at thermoneutrality, male C57BL/6J mice aged 12 weeks were individually housed for 48 h at 28 °C, 22 °C or 4 °C. Each group was given AL access to chow diet throughout. Mice were then euthanised for tissue isolation.

**Mouse PET/CT analysis.** PET/CT scan images were reconstructed and data was analysed using PMOD version 3.806 (PMOD, Zurich, Switzerland). Standardised uptake values (SUV) were calculated for regions of interest, namely BAT, iWAT, gWAT; heart; bone tissue (without BM) from tibiae, femurs, and humeri; and the BM cavities within these bones. To distinguish bone tissue from BM, a calibration curve was generated using HU obtained from the acquisition of a tissue equivalent material (TEM) phantom (CIRS, model 091) and mouse CT scans. The TEM phantom consists of 2–4 mm hydroxyapatite rods representing mass densities of 1.08–1.57 g/mL. The TEM-reconstructed CT image data was exported for analysis into PMOD and, for the extraction of TEM HU values, a VOI template was generated and placed on each rod (0.008 mL for 2 mm and 0.05 mL 4 mm). The calibration curve was plotted based on the calculated linear equation of the TEM HU values, in which the mouse tissue values were inserted/scaled. This ensured that, within whole bones, regions of interest were specific for bone or BM.

**Human PET/CT studies.** Subjects with active BAT at room temperature were identified retrospectively from clinical PET/CT scans. While it is unclear why some people have active BAT at room temperature, this observation is consistent with previous reports showing that 7.5% of females and 3.1% of males have detectable BAT activity at room temperature[63]. A control group, without detectable BAT, was then identified, ensuring that age, sex, weight, BMI and fasting blood glucose were matched to the active-BAT group (Table 1).

To assess effects of cold exposure (Fig. 7, Supplementary Fig. 6A, B) or prednisolone treatment (Supplementary Fig. 6C), subjects attended the Clinical Research Facility at the Royal Infirmary of Edinburgh in standard light clothing after an overnight fast; at each visit they wore identical clothing and were instructed to avoid alcohol or exercise for the preceding 48 h. For each study, subjects were first placed in a room at 23–24 °C (warm room) and measurements were made of height, weight and body fat using bioimpedance (using an Omron BF-302). For cold exposure[8,53], subjects were then placed in a room at 17 °C (cold room) for 2 h to activate BAT. Subjects were asked and assessed clinically for shivering every 15 min; no shivering was detected clinically or noted by any subject during cold exposure. After 1 h of cold exposure, subjects received an intravenous injection of 185 MBq ¹⁸FDG. Subjects were then placed prone in a hybrid PET/CT scanner (Biograph mCT, Siemens Medical Systems). The PET scan commenced 1 h post-¹⁸FDG injection after an initial low-dose CT for attenuation correction (non-enhanced, 120 kV). Static PET imaging of the upper body was then done using 5-min beds and images were analysed using PMOD version 3.806 (PMOD, Zurich, Switzerland).

To assess effects of prednisolone treatment, subjects were recruited to a double-blind, randomised crossover study[8]. Subjects were randomised to receive three doses of 10 mg prednisolone or placebo 12 h apart prior to each study visit (at 0800 h and 2000 h the day prior to each study visit; and at 0800 h on the morning of the study visit). The first and second visits were held at least 2 weeks apart to allow sufficient washout. During each visit subjects in the warm room received an infusion of 6,6-[²H]₂-glucose, commencing at 0.22 μmol/kg/min for 180 min following an initial bolus of 17.6 μmol/kg. Subjects then remained at rest in the warm room for 2 h. Thereafter, subjects were transferred to the Clinical Research Imaging Centre and were placed supine in a room cooled to 17 °C (cold room) for 2 h. Subjects were checked every 15 min for signs or symptoms of shivering. Following 1 h in the cold room, subjects were given an intravenous injection of 75 MBq ¹⁸FDG. PET scanning at 1 h post-¹⁸FDG injection was then done as for the cold-exposure study, with the exception that subjects lay supine in the scanner, with scans performed using 10-min beds.

To estimate the total glucose uptake capacity of BMAT, bone, RM and skeletal muscle, the total-body volume of each of these tissues was first estimated for the subjects in the No BAT and Active BAT groups (Figs. 7 and S6). BMAT and RM volumes were estimated based on the observations that BM accounts for approximately 5% of body mass[64]; that, in adult humans, BMAT accounts for ~70% and RM ~30% of BM[65]; and that fat and haematopoietic tissues have densities of 0.92 and 1.06 g/mL, respectively[66]. Bone mass was estimated based on the height of each subject[67] and from this the volume of bone was calculated based on a density of 1.245 g/mL[66]. Skeletal muscle mass was estimated based on reference ranges (skeletal muscle as % body mass) reported by Janssen et al.[68] using values relevant to the age and sex of each subject. From these estimated masses, skeletal muscle volume was then calculated based on a density of 1.055 g/mL[69]. Thus, based on the body mass or height, age and sex of each subject, we estimated the total volume (mean ± SD, L) of each tissue as follows: BMAT, 2.27 ± 0.37; RM, 0.84 ± 0.14; bone, 7.51 ± 0.82; and skeletal muscle, 17.93 ± 5.34. Total FDG uptake was then estimated by multiplying these volumes by the SUV for each tissue (Fig. 7c, d). Skeletal muscle SUVs were based on the average SUVs in leg muscles, biceps and triceps. Total uptake in BMAT, RM or bone was calculated from the average SUVs for these sites in the axial or appendicular skeleton.

**Determination of attenuation density for BMAT in humans.** HU of subcutaneous fat, yellow marrow and RM were determined using Analyse 12.0 software (AnalyzeDirect, Overland Park, KS, USA) based on data from 33 patients who had undergone paired CT and MRI scans (Table 1). The MRI sequence was an axial HASTE (Half Acquisition Single Shot Turbo Spin Echo) with a TE (echo time) of 50 ms, TR (repetition time) of 1000 ms, and slice thickness 8 mm. In

HASTE MR techniques, BM fat corresponds to higher signal intensity compared to surrounding bone and muscle tissues. CT scanning, as described previously[70], was performed using a 320-multidetector scanner (Aquilion ONE, Toshiba Medical Sytems, Japan). After acquisition of scout images, patients underwent non-contrast wide-volume CT using tube voltage 120 kV, tube current based on BMI and scan range from 2 cm below the carina to the base of the heart using volume sizes of 160, 140, 128, 120, 100 or 80 mm. Using Analyse 12.0 software, the MR and CT scans were co-registered and VOI were manually drawn around the sternum, vertebrae and subcutaneous AT. HU were extracted on a per voxel basis, and data underwent post-processing using a customised in-house software, developed in Matlab, to measure the total number of voxels across all patient HU (Fig. 6b). ROC analysis of the CT data (MedCalc) was then conducted on per-voxel HU to determine threshold values with the greatest sensitivity and specificity to detect bone, yellow marrow and RM. Thresholds of above 300 HU were defined as bone regions, −200 to 115 HU as yellow marrow, and 115–300 as RM.

**Micro-CT scanning (μCT).** Following euthanasia, murine tibiae were isolated, thoroughly cleaned and fixed in 10% formalin at 4 °C for 48 h. Bones were decalcified for 14 days in 14% EDTA and washed in Sorensen's phosphate buffer. Bones were then stained for 48 h in 1% osmium tetroxide (Agar Scientific, UK), washed in Sorensen's phosphate buffer and embedded in 1% agarose, forming layers of five tibiae arranged in parallel in a 30-mL universal tube. Tubes of embedded tibiae were then mounted in a Skyscan 1172 desktop micro CT (Bruker microCT, Kontich, Belgium). Samples were scanned through 360° using a step of 0.40° between exposures. A voxel resolution of 12.05 μm was obtained in the scans using the following control settings: 54 kV source voltage, 185 μA source current with an exposure time of 885 ms. A 0.5 mm aluminium filter and two-frame averaging were used to optimise the scan. After scanning, the data were reconstructed using NRecon v1.6.9.4 software (Bruker, Kontich, Belgium). The reconstruction thresholding window was optimised to encapsulate the target image. Volumetric analysis was performed using CT Analyser v1.13.5.1 (Bruker microCT, Kontich, Belgium).

**Statistical analysis, data presentation and reproducibility.** Microarray data were analysed as described above. All other data were analysed for normal distribution within each experimental group using the Shapiro–Wilk normality test. Normally distributed data were analysed by ANOVA or *t* tests, as appropriate. Where data were not normally distributed, non-parametric tests were used. When appropriate, *P* values were adjusted for multiple comparisons. Data are presented as histograms or box and whisker plots. For the latter, boxes indicate the 25th and 75th percentiles; whiskers display the range; and horizontal lines in each box represent the median. All statistical analyses were performed using Prism software (GraphPad, USA). A *P* value < 0.05 was considered statistically significant. Units and abbreviations are reported in accordance with recently published guidelines for research relating to BM adiposity[71]. Where representative micrographs or PET/CT images are shown, figure legends describe the number of biologically independent samples that these are representative of.

**Reporting summary.** Further information on research design is available in the Nature Research Reporting Summary linked to this article.

## Data availability

All relevant data are available from the authors upon reasonable request. The source data underlying Figs. 1b–e, g, 2a, b, 3a, c, e, f, 4a, b, 5a, c, e–g, 6b, e, 7b–f, S2C, D, S3B–E, S4A–D, F, S5A–D and S6B–C are provided as a Source Data file. Microarray data for analysis of rabbit cohorts 1 and 2 has been made publicly available on the NCBI GEO platform[72] under the series ID GSE138690. The Molecular Signature Database (MSigDB, v6.2), used for Gene Set Enrichment Analysis, can be accessed at https://www.gsea-msigdb.org/gsea/msigdb/collections.jsp Source data are provided with this paper.

## Code availability

Code for the in-house ROC analysis software has been deposited in GitHub and is available at https://github.com/Georgerun/ROCPerPixel. Source data are provided with this paper.

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

## Acknowledgements

This work was supported by grants from the Medical Research Council (MR/M021394/1 to W.P.C.; MR/K010271/1 to R.H.S.), the National Institutes of Health (R01 DK62876 and R24 DK092759 to O.A.M.; K99-DE024178 to E.L.S.; and P30 DK089503 to the Michigan Nutrition Obesity Research Center), the Wellcome Trust-University of Edinburgh Institutional Strategic Support Fund (to W.P.C. and K.J.S.), and the British Heart Foundation (4-year BHF PhD Studentship to R.J.S., B.J.T., M.C.S. and B.M.N; BHF CoRE Bioinformatics Grant to W.P.C.; BHF CoRE grant to A.J.D.). W.P.C. is further supported by a Chancellor's Fellowship from the University of Edinburgh. A.A.S.T was funded by the British Heart Foundation (RG/16/10/32375). E.J.R.v.B is supported by SINAPSE (the Scottish Imaging Network). We are grateful to the British Heart Foundation for providing funding towards establishment of the Edinburgh Preclinical PET/CT laboratory (RE/13/3/30183), and to NHS Research Scotland (NRS) for financial support of the Edinburgh Clinical Research Facility. R.H.S and M.C.W. are supported by The Chief Scientist Office of the Scottish Government (SCAF/17/02 to R.H.S.; PCL/17/04 to M.C.W.). J.P.M.A is supported by BHF Clinical Research Training Fellowship no. FS/17/51/33096. We are grateful to John Henderson (BVS, University of Edinburgh) for support with mouse husbandry; Anish K. Amin (Department of Orthopaedic Surgery, Royal Infirmary Edinburgh), Beena Polouse and Frank Morrow (Edinburgh Clinical Research Facility,) for help with human studies; Tashfeen Walton and Christophe Lucatelli (Edinburgh Imaging, University of Edinburgh) for radiotracer production; Robert K. Semple (Centre for Cardiovascular Science, University of Edinburgh) for critical feedback on this paper; and to staff at the University of Michigan microarray core facility for processing of rabbit microarray data.

## Author contributions

Conceptualisation: K.J.S. and W.P.C.; Methodology: K.J.S., A.A.S.T, E.L.S., G.P., C.G., L.E.R., W.A.M., A.J.D., A.P., S.C., R.J.W., O.A.M., N.M.M, R.H.S. and W.P.C.; Investigation: K.J.S., D.M., E.L.S., L.E.R., G.P., C.G., A.L., R.J.S., B.J.T., M.C.S., B.M.N., C.J.A., D.S., G.J.M., M.R.D., J.P.M.A., M.C.W., R.J.W., E.J.R.v.b., R.H.S. and W.P.C.; Formal analysis: K.J.S., A.A.S.T, D.M., E.L.S, G.P., C.G., W.A.M., J.P.M.A., M.C.W., R.J.W., E.J.R.v.B., N.M.M., R.H.S. and W.P.C.; Resources: W.P.C., A.A.S.T., A.P., S.C., G.J.M., M.R.D., J.P.M.A., M.C.W., E.J.R.v.b., N.M.M. and R.H.S.; Writing—Original Draft: K.J.S. and W.P.C.; Writing—Review & Editing: K.J.S., A.A.S.T, E.L.S., G.P., W.A.M., A.J.D., S.C., M.C.W., R.J.W., E.J.R.v.b., O.A.M., R.H.S. and W.P.C.; Visualisation: K.J.S. and W.P.C.; Supervision: A.J.D., A.P., M.R.D., O.A.M., N.M.M., R.H.S. and W.P.C.; Funding acquisition: A.J.D., O.A.M., N.M.M., R.H.S. and W.P.C.

## Competing interests

E.J.R.v.B. has received the research support from the Siemens Healthineers and is the owner of QCTIS Ltd. Remaining authors declare no competing interests.
