## [Peer Review File · Nature Communications]

Reviewers' Comments:

Reviewer #1:

Remarks to the Author:

This work from the Cawthorn group explores the hypothesis that BMAT, bone marrow adipose tissue, is a distinct type of adipose tissue, using rabbit models (via whole rabbit tissue transcriptomic analysis), and human analyses. The data are crucial to the field, as the field had no clear evidence that BMAT was distinct from white adipose tissue (WAT), brown adipose tissue (BAT) or beige adipocytes at the gene expression level. Moreover, this is the first study to demonstrate that BMAT does not respond to insulin by uptaking glucose as much as other adipose depots. Importantly, they also demonstrate that BMAT is not beige-like/brown, because it does not respond to cold exposure or glucocorticoids in the same way that brown fat does. This will hopefully put to rest the misunderstandings in the field about whether or not BMAT is a type of brown/beige adipose. A few questions should be addressed before acceptance:

More details on how whole BMAT isolated from the proximal tibia (pBMAT), distal tibia (dBMAT) or radius and ulna (ruBMAT) of two cohorts of rabbits would be useful. Were these samples pure adipocytes, or were other cells included in these samples? Was sorting of cells done? These details are in the methods of the papers they reference, but would be useful to reproduce here with any modifications this group made. Images to show purity/characterization of adipocytes would be helpful.

Same question for human adipocytes- what type of purity was obtained? Did you obtain 100% pure adipocytes? The writing is unclear in the results what the difference is between "human femoral BMAT" and "adipocytes isolated from trabecular bone". The reader would have to read the Craft 2018 paper and the Mattuucci et al paper to understand what "adipocytes from trabecular bone" are as compared to "human femoral BMAT"- at least an overview of the differences should be written into this manuscript.

They state that "compared to WAT, BM and BMAT resist insulin-stimulated glucose uptake". This appears to be true, and is very interesting, but can they explain why insulin in fact appears to decrease glucose uptake in the humerus? There appears to be very little BMAT there, as seen in Fig 2E, so is that really a legitimate BMAT to quantify?

Can the authors speculate as to what makes adipocytes uptake glucose in the BM, if it is not via insulin? Also, the authors show there is high basal metabolic activity in BMAd, and that they contain lots of mitochondria, but can they explain what they need this energy for? If these cells are just storing lipids, why do they need so much glucose?

Are there in vitro studies in seahorse that show that BMAd more than white adipocytes take up glucose? Do they utilize it differently? Do they uptake less of another energy source (protein or fat) than white adipocyte? IE. Is there total caloric intake different or just the source of energy?

Supp Fig S3B: What is H4(am)? That is energy expenditure unit? Please write it out, and explain it and say what am means. Also, please state why it is higher in acute cold- since you only explain that the chronic cold increased energy expenditure is due to increased food intake. Is this due to shivering?

Can you explain why some of your patients "active BAT at room temperature" while other did not. Fig 4E- I am surprised there is a significant difference between RM and BMAT in the sternum - this does not look significant. How were these stats performed- were all 3 groups merged? The statistics are a bit misleading as it looks like the authors are just saying there is significance between the No BAT groups for these areas- but I don't think that is the correct message. The same questions apply to the comparison between Bone and BMAT in the sternum of Fig 4E.

Minor comments to authors:

The wording should be Distinct from, not Distinct to.

PET images should have scale bars if possible. Why does the left side of the mouse in Figure 3A (mouse 3) have such higher signal than the right side?

- Michaela Reagan

Reviewer #2:

Remarks to the Author:

This short article reported the unique characteristics of bone marrow adipose tissue (BMAT) and its role in systemic glucose homeostasis. The authors used transcriptomics to analyze the profiles of different fat depots and revealed that BMAT is molecularly distinct from either white, brown or beige adipose tissues. BMAT displayed greater basal glucose uptake but is resistant to both insulin- and cold-stimulated glucose uptake, which further highlights the functional distinction between BMAT and white or brown adipose tissues. The authors also established a PET/CT method for BMAT localization and functional analysis of glucose uptake in both mice and humans. Overall, the data are convincing and supporting the major conclusions. These findings are novel and provide new insights into the potential functions of BMAT. Moreover, the new method for BMAT identification from clinical CT scans established in this paper could be applied in other clinical and preclinical studies, which will extend further investigations of BMAT functions. Here are a few specific comments:

1. For Figure 1, what is the rationale to use rabbit cohorts rather than mouse cohorts for transcriptional profiling? Are these findings also applicable in mice (as you examine insulin- and cold-induced glucose uptake in mice)?
2. Your conclusion of Figure 2 is that insulin does not induce glucose uptake in BMAT, I therefore wonder if you could provide the protein expression of insulin receptor in BMAT as well. Furthermore, you need to add an in vitro experiment by treating isolated BMAT with insulin and checking glucose uptake in these cells to further confirm your conclusion.
3. In Figure 3, you show that BMAT is resistant to cold-stimulated glucose uptake. Since cold exposure also induces lipolysis, I'm wondering whether the lipolytic gene expression profile of BMAT after cold challenge is altered or not.
4. There is one publication (Erica L. Scheller, et al. Region-specific variation in the properties of skeletal adipocytes reveals regulated and constitutive marrow adipose tissues. Nature Communications, 2015) showing that BMAT in proximal tibia was dramatically reduced after 21-day cold exposure. In Fig. 3B, the authors showed that glucose uptake is unchanged in proximal tibia but decreased in distal tibia after 72-hour cold challenge. Considering the findings of the previous publication, it's also necessary to show whether the number/abundance of BMAT in different regions is changed after 72-hour cold, preferably using osmium staining. This will further explain whether the unchanged or decreased cold-induced glucose uptake in BMAT is due to altered BMAT abundance or BMAT is resistant to cold-induced glucose uptake indeed.
5. Similar to Comment 4, in addition to glucose uptake, it's also necessary to show the CT images highlighting BMAT in humans at room temperature and cold environment in Figure 4, which will further consolidate your conclusion.
6. Could you explain which type of glucose transporter is responsible for basal glucose uptake in BMAT?
7. What is the contribution of BMAT-mediated basal glucose uptake to systemic glucose homeostasis?

Other minor issues:

1. The resolution of PET/CT images (Fig. 2B, Fig. 2D, Fig. 3A) is too low. You should show images with higher resolution.

2. For Fig. 3D, you'd better use immunohistochemical staining of UCP1 rather than H&E staining. In addition, the image of iWAT after chronic cold exposure (Fig. 3D) didn't show obvious beige phenotype compared to acute cold and control groups. You need to show a more representative one.

Reviewer #3:

Remarks to the Author:

In "Bone marrow adipose tissue is a unique adipose subtype with distinct roles in systemic glucose homeostasis" by Suchacki et al., the authors address fundamental questions related to the physiological relevance of bone marrow adipose tissue (BMAT). They first demonstrated in mouse tissue that BMAT has a different transcriptional profile compared to either white or brown and beige/brite tissues. Curiously, neither insulin nor cold exposure increased glucose uptake in BMAT, as assessed by both adipose tissue biopsies and 18F-FDG PET/CT, the latter being a novel approach to assessing an adipose tissue not easily accessible. They finally turned to human BMAT using 18F-FDG PET/CT and report that in the basal state, BMAT is a major site of glucose uptake.

The manuscript has several strengths, among which are the excellent execution of technically challenging experiments; the introduction of a method to study human BMAT in situ via non-invasive imaging; and the use of different animal and human models of BMAT to create a fuller picture of its physiological roles. It is striking that BMAT plays little role in glucose uptake in response to insulin, cold exposure, or even glucocorticoids, the last of which the authors described in previous publications.

The manuscript could be improved by addressing the following points:

1. Physiological significance of BMAT in rodents and humans.

The authors appropriately support the part of the title stating that BMAT is a unique adipose tissue. What is not clear is how it could influence systemic glucose homeostasis. Much of the data provided actually suggest it is quite metabolically inert in that it does not respond to insulin, cold exposure, or glucocorticoids. It sounds as if the authors were unsatisfied with the extent of the advances they made in the understanding of BMAT physiology [nicely discussed in the first paragraph of the Discussion and other locations as well]. Instead, they concluded with a rather broad claim that BMAT has the "potential to influence systemic glucose homeostasis."

1A. Quantification

Figs. 4F and 4G are critical to the paper vis-à-vis the human BMAT. The glucose uptake, as measured by SUV, is higher in sternum compared to bone and red marrow in 3 of the 5 sites studied; and glucose uptake in BMAT is higher than subcutaneous WAT. However, these are all very low SUV values. The authors report that BMAT can be >10% of total adipose tissue mass. They now need to show in grams of glucose how much glucose is taken up by the bone, RM, scWAT, and BMAT sites in the basal state. These values should be compared to skeletal muscle. It would be helpful if skeletal muscle SUV was shown in Fig. 2F.

The authors can then discuss these absolute numbers and determine whether the BMAT depots represent a major site for glucose uptake in the basal state, both compared to other WAT depots as well as skeletal muscle. Either substantial or not, BMAT's glucose uptake amounts would be of importance.

1B. How is lipogenesis in BMAT regulated if it is not insulin responsive?

1C. What could the physiological roles of BMAT be if it does not respond to critical hormonal regulators such as insulin?

2. Clinical trial information

2A. Was this clinical trial registered with any central database such as the EU Clinical Trials Register?

2B. Was the number of human subjects studied prespecified? Was a sample size calculation or at least a detectable difference calculation done?

Minor comments:

In graphs where mRNA expression is lower than 1.0, it would be helpful to use a log scale. Examples are 1G, 3E-G, and S3F-I.

In 3B, it is not clear what the highest value is before the axis break. Is it 2.0?

The phrase should be "distinct from" not "distinct to."

Reviewer #1

This work from the Cawthorn group explores the hypothesis that BMAT, bone marrow adipose tissue, is a distinct type of adipose tissue, using rabbit models (via whole rabbit tissue transcriptomic analysis), and human analyses. The data are crucial to the field, as the field had no clear evidence that BMAT was distinct from white adipose tissue (WAT), brown adipose tissue (BAT) or beige adipocytes at the gene expression level. Moreover, this is the first study to demonstrate that BMAT does not respond to insulin by uptaking glucose as much as other adipose depots. Importantly, they also demonstrate that BMAT is not beige-like/brown, because it does not respond to cold exposure or glucocorticoids in the same way that brown fat does. This will hopefully put to rest the misunderstandings in the field about whether or not BMAT is a type of brown/beige adipose. A few questions should be addressed before acceptance:

1.1 More details on how whole BMAT isolated from the proximal tibia (pBMAT), distal tibia (dBMAT) or radius and ulna (ruBMAT) of two cohorts of rabbits would be useful. Were these samples pure adipocytes, or were other cells included in these samples? Was sorting of cells done? These details are in the methods of the papers they reference, but would be useful to reproduce here with any modifications this group made. Images to show purity/characterization of adipocytes would be helpful.

Thank you for these very useful suggestions. We have now updated the Results (Lines 140-148) and Methods (Lines 679-688) to provide further details of the rabbit BMAT isolation. We have also updated Figure S1 to show representative micrographs of the isolated tissues (now in Fig. S1A). These samples were whole BMAT, without isolating or sorting cells, as shown in the micrographs. One benefit to using rabbits is that very pure pieces of cBMAT can be dissected from the distal tibia and forearms, without contamination with red marrow. This purity is supported by the fact that the microarrays for whole rabbit BMAT vs WAT give very similar results to the microarrays for isolated BMAds and white adipocytes from humans. In contrast, the rabbit tibial pBMAT does contain red marrow (Fig. S1A), which is why this depot was not included in the subsequent differential expression analyses (i.e. Figs. 1B-D and S1B-C).

1.2 Same question for human adipocytes- what type of purity was obtained? Did you obtain 100% pure adipocytes? The writing is unclear in the results what the difference is between “human femoral BMAT” and “adipocytes isolated from trabecular bone”. The reader would have to read the Craft 2018 paper and the Mattiucci et al paper to understand what “adipocytes from trabecular bone” are as compared to “human femoral BMAT”- at least an overview of the differences should be written into this manuscript.

For the microarrays of human adipocytes, purity was confirmed by confocal microscopy as reported in Mattiucci *et al*, 2018 (Mattiucci et al., 2018). We took a similar approach for adipocytes isolated from human cohort 2 (Fig. 1G, S2C-D): aliquots of the isolated adipocytes were stained with Hoescht and LipidTox Deep Red, followed by analysis using a Nexcelom cell counter to assess successful isolation of nucleated adipocytes. Example images are shown in Figure R1 of our response to reviewers. Here, you can see that each nucleus is associated with a large unilocular lipid droplet. There are some lipid droplets that are not colocalised with a nucleus, perhaps because their nuclei are on the opposite side of the cell and so are not visible. The brightfield images reveal some non-nucleated debris, but also show the expected morphology of the isolated adipocytes. We could not do this analysis on all of our adipocyte samples, particularly those isolated from trabecular bone, because these often gave low cellular yields and we therefore prioritised these samples for RNA isolation. Another limitation is that the Nexcelom images are quite low-resolution (this instrument is primarily for cell counting rather than high-resolution imaging). For these reasons we decided to not include these images in the manuscript. However,

based on these we are confident that our method has been successful in isolating pure adipocytes.

To further assess adipocyte purity, we have now used qPCR to analyse the expression of CD45 (*PTPRC*) and CD11B (*ITGAM*) as markers of haematopoietic cells and macrophages, respectively. This new data is shown in Fig. S2C. Relative to the SVF, BMAds and Bone Ads have greater expression of these markers than do WAT Ads, which likely reflects the much greater proportion of haematopoietic and immune cell types within the BM than within WAT. Importantly, in each tissue the expression of these two transcripts is significantly higher in the stromal vascular fraction (SVF) than in the adipocyte fraction, which further validates our method for isolating adipocytes from each of these tissues. We have now updated the Results section of the manuscript (Lines 170-173) to describe these new results.

Finally, we have updated the text (lines 166-169) to clarify the differences between the femoral BMAds and those from trabecular bone, writing “*White adipocytes were obtained from subcutaneous gluteofemoral WAT, while BMAds were isolated from the proximal femoral diaphysis; we also isolated adipocytes from trabecular bone of the proximal femoral metaphysis (Bone Ads) to assess potential site-specific differences in BMAd function*”⁹.

1.3 They state that “compared to WAT, BM and BMAT resist insulin-stimulated glucose uptake”. This appears to be true, and is very interesting, but can they explain why insulin in fact appears to decrease glucose uptake in the humerus? There appears to be very little BMAT there, as seen in Fig 2E, so is that really a legitimate BMAT to quantify?

We agree that this effect is very interesting but currently cannot explain its basis. We also agree that the humerus BM does not give information about the response of BMAT, because, at this age, it contains no BMAT; this is actually one reason why we included the humerus, because we wanted to assess responses of different regions of the bone and BM, including those that are BMAT-deficient. We apologise if we did not make this clear in the text and so have now updated the text of the Results section (Lines 214-215) to include the statement, “*Surprisingly, insulin was associated with decreased glucose uptake in the humerus BM, which is BMAT-deficient*”. We also address this in Lines 372-376 of the Discussion, stating, “*It is unclear why insulin is associated with decreased glucose uptake in humeral bone and BM; this is unlikely to be a technical issue given that we see the expected insulin-stimulated glucose uptake in the heart, skeletal muscle, WAT and femur. Thus, the lack of increases in humeri and tibiae suggests that there are site-specific differences in skeletal insulin responsiveness*”.

1.4 Can the authors speculate as to what makes adipocytes uptake glucose in the BM, if it is not via insulin? Also, the authors show there is high basal metabolic activity in BMAds, and that they contain lots of mitochondria, but can they explain what they need this energy for? If these cells are just storing lipids, why do they need so much glucose?

We speculate that the high glucose uptake in BMAds is due to their relatively high expression of SLC2A1 and SLC2A3; we emphasise this point in lines 463-470 of the Discussion.

Functionally, we think that BMAds are not just storing the lipids but are using these to provide fatty acids as an essential fuel source to support haematopoiesis. The high basal glucose uptake is likely required for *de novo* lipogenesis, to ensure that BMAds have sufficient triglyceride reserves. We have now updated the Discussion (Lines 475-485) to emphasise these important points, stating, “*This high level of basal glucose uptake is also likely to be important for BMAd function. Ex vivo studies show that glucose incorporation into triglycerides is greater for BMAT than for*

gWAT⁴⁶, while whole BM displays very high rates of de novo lipogenesis from ¹⁴C-glucose in vivo⁴⁷. This suggests that high basal glucose uptake can make a substantial contribution to lipogenesis in BMAT. Moreover, BMAd-derived fatty acids fuel the proliferation and survival of haematological tumours⁴⁸, and fatty acid oxidation is important for maintenance of haematopoietic stem cells⁴⁹. Based on these observations, we speculate that BMAds' high level of basal glucose uptake is required for de novo lipogenesis and the ability of BMAds to act as a local energy source to support haematopoiesis. The relative insulin resistance of BMAds may then allow them to maintain this essential function even in the face of systemic energy deficiency”.

1.5 Are there in vitro studies in seahorse that show that BMAds more than white adipocytes take up glucose? Do they utilize it differently? Do they uptake less of another energy source (protein or fat) than white adipocyte? IE. Is there total caloric intake different or just the source of energy?

These studies have not yet been done in the Seahorse analyser, but we agree that this such respirometry would be extremely informative and should be pursued in future studies. One challenge is that BMAds differentiated *in vitro* from BMSCs do not recapitulate the properties of primary BMAds (Attané et al., 2019), while for primary adipocytes collagenase digestion can influence intracellular signalling pathways (Engfeldt et al., 1980). Therefore, any Seahorse analyses would have to be carefully optimised for use of primary BMAds.

In terms of glucose utilisation and uptake of other energy sources, there are some data in the literature but many questions remain unanswered. Several older studies have assessed BMAT glucose uptake *in vivo* or using explants of BMAT. For example, in 1967 Zakaria and Shafir reported that incorporation of glucose into triglycerides is greater for explants of BMAT than explants of gWAT (Zakaria and Shafir, 1967). In 1988 Kozubik *et al* used ¹⁴C-glucose to assess the effects of fasting and refeeding on *de novo* lipogenesis in rats. They found that, compared to fasted animals, refeeding increases lipogenesis in WAT, liver and bone marrow, but both the absolute rates of lipogenesis are greatest for the bone marrow (Kozubik et al., 1988). Unlike our studies, these previous experiments do not specifically assess BMAT glucose uptake *in vivo*; however, they suggest that BMAds have high glucose uptake to support lipogenesis. Studies from the 1970s show that rates of fatty acid uptake and esterification are also higher in BMAT than in WAT (Bathija et al., 1979; Trubowitz and Bathija, 1977), while uptake of polyunsaturated fatty acids was recently found to be greater in the skeleton than in gWAT of mice (Bartelt et al., 2017). To our knowledge, BMAT's use of protein as an energy source has not been assessed.

In the revised manuscript we now address these issues in lines 470-473 of the Discussion by stating, “*Uptake and esterification of fatty acids is also greater in BMAT than in WAT^{38,45}; however, the relative contribution of amino acids to BMAd energy requirements remains to be established. Future studies, including the use of respirometry, may prove extremely helpful in further dissecting these aspects of BMAd bioenergetics”.*

1.6 Supp Fig S3B: What is H4 (am)? That is energy expenditure unit? Please write it out, and explain it and say what am means. Also, the please state why it is higher in acute cold- since you only explain that the chronic cold increased energy expenditure is due to increased food intake. Is this due to shivering?

Sorry for not explaining this more clearly. We have now updated the legend to Figure S3 to state that H4 “... is calculated by measuring the energy produced (H3, Watts) during the light cycle (7 am to 7 pm) and dividing this by the mouse's lean body mass (kg)”. The y-axis label has also been updated to state the units of H4 (W/kg). We have also updated the text of the Results section (lines 238-240) as follows: “As shown in Fig. S3B, acute (4 h) or chronic cold (72 h) increased

energy expenditure (H4), reflecting the increased metabolic rate required to maintain body temperature". In addition, we have analysed ^{18}F -FDG uptake into skeletal muscle and find no increase in the acute or chronic cold mice. Therefore, in the text (lines 242-245) we now explain, "Acute or chronic cold exposure increased BAT ^{18}F -FDG uptake but did not increase uptake into skeletal muscle (Fig. 5A-C), indicating that non-shivering thermogenesis, rather than shivering, was likely responsible for the increased energy expenditure during cold exposure".

These text changes clarify that H4 is a measure of energy expenditure and explain that these increases result from non-shivering thermogenesis. This is apparent from the marked induction of *Ucp1* in BAT and iWAT of acute and chronic cold mice (Fig. 5E-F; Figures 3E-F in our original submission). For the chronic cold mice we don't think that the increased food intake is contributing to the increased energy expenditure, but rather is a result of the increased energy expenditure to prevent weight loss and hypoglycaemia during chronic cold.

1.7 Can you explain why some of your patients "active BAT at room temperature" while other did not.

Subjects with active BAT at room temperature were identified retrospectively from clinical PET/CT scans. It is unclear why some subjects have active BAT whereas others do not, but this observation of variability in basal BAT activation is consistent with previous studies. For example, Cypess *et al* reported that, in regions extending from the anterior neck to the thorax, detectable active BAT was seen in 7.5% women and 3.1% men (Cypess *et al.*, 2009). To clarify this point we have now updated the methods (Lines 905-909) to state, "While it is unclear why some people have active BAT at room temperature, this observation is consistent with previous reports showing that 7.5% females and 3.1% males have detectable BAT activity at room temperature⁶²; a control group, without detectable BAT, was then identified, ensuring that age, sex, weight, BMI and fasting blood glucose were matched to the active-BAT group".

1.8 Fig 4E- I am surprised there is a significant difference between RM and BMAT in the sternum - this does not look significant. How were these stats performed- were all 3 groups merged? The statistics are a bit misleading as it looks like the authors are just saying there is significance between the No BAT groups for these areas- but I don't think that is the correct message. The same questions apply to the comparison between Bone and BMAT in the sternum of Fig 4E.

The data in Fig.4E (Fig. 7B of revised manuscript) is significant because, by paired analysis (within each subject), the sternum BMAT has higher glucose uptake than the sternum RM in 12 of 15 patients analysed; this is presented in Figure 7E (Fig. 4F of original submission). In 4E (now Fig. 7B) the data for the sternum were assessed by paired 2-way ANOVA, because the tissues are matched (paired) between each subject and because two independent variables are being assessed (i.e. skeletal site [Bone, RM, BMAT] and subject group [no BAT, active BAT, or cold]). In Figure 7E (previously 4F) the data for each bone are assessed by one-way ANOVA, because for this data the only independent variable is the tissue type within each bone (i.e. Bone, RM, BMAT). Please note that for 7E we have combined the 'no BAT' and 'active BAT' subjects, because analysis in Fig. 7B-D (previously 4E and S4D) shows that there are no differences in glucose uptake between the 'no BAT' and 'active BAT' groups. Therefore, combining these two groups (in Fig. 7E) provides more power for detecting significant differences in glucose uptake between the different tissue types. This also explains why the degree of significance (between BMAT, bone and RM) is greater in Fig. 7E than in Fig. 7B-D. We have updated the legend for this Figure (now Fig. 7) to explain, stating, "Data in (B-D) are mean \pm SEM of 8 (No BAT) or 7 (Active BAT, Cold) subjects per group and were analysed by paired two-way ANOVA. Data in (E-F) are shown as paired individual values (E) or box-and-whisker plots (F) of subjects in the No BAT and Active BAT

groups and were analysed by paired one-way ANOVA. For (B-E), significant differences between bone, RM and BMAT are indicated by * ($P < 0.05$), ** ($P < 0.01$) or *** ($P < 0.001$). Significant differences between No BAT, Active BAT and Cold groups are indicated by # ($P < 0.05$) or ### ($P < 0.001$). For (F), tissues that do not share a common letter have significantly different SUVs”.

Minor comments to authors:

1.9 The wording should be Distinct from, not Distinct to.

Thanks, we have now corrected this in the text (highlighted in manuscript).

1.10 PET images should have scale bars if possible. Why does the left side of the mouse in Figure 3A (mouse 3) have such higher signal than the right side?

Thank you for this suggestion, we have now added scale bars to all of the PET images to show how the colours in the images relate to the SUVs (in g/mL). For the mouse in Figure 3A (Figure 5A in revised manuscript) the PET signal on the left-hand side is from FDG uptake in paraspinal BAT. Mice typically exhibit hunching during cold exposure and this can skew their spinal column to one side. To clarify this in the revised manuscript we have updated the legend to Figure 5 and updated the images in Figure 5A to show the full spinal column.

Reviewer #2

This short article reported the unique characteristics of bone marrow adipose tissue (BMAT) and its role in systemic glucose homeostasis. The authors used transcriptomics to analyze the profiles of different fat depots and revealed that BMAT is molecularly distinct from either white, brown or beige adipose tissues. BMAT displayed greater basal glucose uptake but is resistant to both insulin- and cold-stimulated glucose uptake, which further highlights the functional distinction between BMAT and white or brown adipose tissues. The authors also established a PET/CT method for BMAT localization and functional analysis of glucose uptake in both mice and humans. Overall, the data are convincing and supporting the major conclusions. These findings are novel and provide new insights into the potential functions of BMAT. Moreover, the new method for BMAT identification from clinical CT scans established in this paper could be applied in other clinical and preclinical studies, which will extend further investigations of BMAT functions. Here are a few specific comments:

2.1 For Figure 1, what is the rationale to use rabbit cohorts rather than mouse cohorts for transcriptional profiling? Are these findings also applicable in mice (as you examine insulin- and cold-induced glucose uptake in mice)?

We have updated the Results section (lines 140-145) to explain the rationale for using rabbits rather than mice: “Rabbits were used because their skeletal and BMAT characteristics share similarities with those of humans. For example, unlike rodents, both rabbit and human bones have haversian canals²³ and display extensive BMAT formation that is concentrated around the central vasculature within the long bones^{24,25}. Rabbits also have the practical benefit of allowing isolation of relatively pure pieces of whole BMAT, which is not possible in mice²⁵”. Our findings in rabbits are consistent with previous microarrays in mice that have analysed BMAd vs white adipocytes (Liu et al., 2011). We address this in the Discussion (Lines 348-353) by stating, “More recent microarrays show that *Slc2a4*, *Insr*, *Irs1* and *Irs2* are lower in BMAd than epididymal white adipocytes of mice³⁵. These data are strikingly consistent with our results for transcript expression

in rabbits and humans (Fig. 1, S1-2) and protein expression in rats (Fig. 2, 4) and further support the conclusion that, compared to WAT, BMAT resists insulin-stimulated glucose uptake". In addition, our rabbit microarray data show that BMAT is insulin resistant and is not BAT- or beige-like, observations that are reproduced in our human microarray data and also functionally in our PET/CT studies in both humans and mice. Therefore, we are confident that the transcriptomic analysis from rabbits is applicable to both rodents and humans.

2.2 Your conclusion of Figure 2 is that insulin does not induce glucose uptake in BMAT, I therefore wonder if you could provide the protein expression of insulin receptor in BMAT as well.

Thank you for this suggestion. We attempted to use immunoblotting to detect the IR, IRS1 and GLUT4 in rabbits but the antibodies for these did not work well in rabbits. Thankfully, we have been able to analyse expression of the IR in BMAT and WAT isolated from rats. We have also analysed expression of IRS1 and GLUT4 in these samples. This data is presented in Figure 2 of the revised manuscript and confirms the microarray data from rabbits and humans, revealing that protein expression of IRS1 and GLUT4 is lower in BMAT than in WAT, and that BMAT also tends to have lower expression of the IR precursor.

2.2.1 Furthermore, you need to add an *in vitro* experiment by treating isolated BMAT with insulin and checking glucose uptake in these cells to further confirm your conclusion.

We first attempted to investigate BMAT insulin responsiveness several years ago by using explants of BMAT and gWAT isolated from rabbits, at a time when we did not have access to preclinical PET/CT. These explants were cultured and treated with or without insulin, before administration of ¹⁴C-deoxyglucose to assess glucose uptake. The results of these studies are shown in Figure R2 of this response to reviewers. From these data we originally concluded that the assays were not working as expected because there was no stimulation of deoxyglucose uptake into the tibial BMAT, and the response in gWAT was also inconsistent (no response in one of the four rabbits analysed). However, in light of the new findings in our manuscript these older results may actually be reflecting BMAT's insulin resistance and increased basal glucose uptake in comparison to WAT. Nevertheless, given the lack of robust insulin-stimulated glucose uptake in gWAT, we are not able to draw reliable conclusions from these analyses. We were also concerned that glucose uptake in these studies may have been influenced by differences in the surface area of the explants, something that is very hard to control for. Another concern with using explants or isolated adipocytes is that technical artefacts may be introduced by the isolation procedure. For example, collagenase digestion can alter signalling pathways in adipocytes (Engfeldt et al., 1980), while explant culture causes the downregulation of adipocyte genes, increased hypoxia and induction of proinflammatory signalling pathways (Gesta et al., 2003). Similarly, recent studies have shown that BMAds differentiated *in vitro* from BMSCs do not recapitulate BMAd function *in vivo* (Attané et al., 2019). For these reasons, we are wary of using isolated adipocytes, adipose tissue explants or *in vitro* differentiated BMAds to draw conclusions about the physiological, *in vivo* functions of BMAT and WAT. The power of PET is that it allows precise characterisation of biological function *in vivo*, in a true physiological context. This is why we used PET/CT to assess BMAT metabolism in mice and humans in a much more robust, reliable and physiologically relevant way.

Nevertheless, we appreciate the importance of further testing the conclusions of our PET/CT data. To do so we have now included additional *in vivo* experiments in rats, in which we have assessed the ability of insulin to stimulate Akt phosphorylation in BMAT and gWAT. This new data, shown in Figure 4 of the revised manuscript, shows that BMAT is markedly resistant to insulin-stimulated phosphorylation of Akt, both for S473 and T308. This resistance likely results from the decreased

expression of the insulin receptor and IRS-1 in BMAT, as shown in our transcriptomic analyses of rabbits and humans (Fig. 1D,G) and protein analyses in rats (Fig. 2 of the revised manuscript).

2.3 In Figure 3, you show that BMAT is resistant to cold-stimulated glucose uptake. Since cold exposure also induces lipolysis, I'm wondering whether the lipolytic gene expression profile of BMAT after cold challenge is altered or not.

To investigate this, we have now analysed expression of *Pnpla2* (encoding adipose triglyceride lipase) and *Lipe* (encoding hormone-sensitive lipase) in these samples and in the BAT and iWAT of the cold-exposed mice. These new data, presented in Figure S5 of the revised manuscript, show that both *Pnpla2* and *Lipe* are significantly induced within bones in response to chronic cold exposure, suggesting increased BMAT lipolysis. In contrast, neither *Pnpla2* nor *Lipe* is induced by cold exposure in BAT or iWAT. This is consistent with previous studies showing that cold-exposure does not always induce the expression of lipolysis-related transcripts in these adipose depots (Rosell et al., 2014; Shore et al., 2013) and that lipolysis within BAT is dispensable for cold-induced thermogenesis (Shin et al., 2017).

Importantly, while doing these new qPCR analyses we identified an error in the qPCR data from our original submission, whereby the bone cDNAs were incorrectly unblinded. We strongly regret this mistake but are grateful to have noticed this during the review process. We now present the correct data in Figures 5G and S5C-D. This confirms that, unlike in BAT and iWAT, cold exposure does not induce the expression of *Ucp1* or *Dio2* in the bone samples. However, in these samples chronic cold does increase expression of *Cpt1b* and *Ppara*, similar to the effects in BAT and iWAT. This suggests that cold exposure increases fatty acid oxidation within bones, even if it does not promote a 'beiging' response.

Despite these apparent increases in lipolysis and FA oxidation, our updated analyses (discussed in response to your next point) show that three-day cold exposure does not decrease BMAd size or overall BM adiposity (Figure S4). This suggests that any increase in lipolysis must be balanced by corresponding increases in lipogenesis. Consistent with this, we found that within bones cold exposure is associated with increased expression of diacylglycerol O-acyltransferase 2 (*Dgat2*), the key enzyme in triacylglycerol synthesis. This may explain why BMAd size does not decrease.

Finally, while doing these new analyses we also measured expression of meteorin-like (*Metrn1*), another marker of cold exposure in BAT and iWAT (Rao et al., 2014). We confirm that cold exposure increases *Metrn1* expression in these adipose depots but not within bones (Figure S5), which further supports our conclusions that BMAT is distinct from BAT and beige adipose tissues.

These new results are described in lines 249-261 and 380-390 of the revised manuscript.

2.4 There is one publication (Erica L. Scheller, et al. Region-specific variation in the properties of skeletal adipocytes reveals regulated and constitutive marrow adipose tissues. Nature Communications, 2015) showing that BMAT in proximal tibia was dramatically reduced after 21-day cold exposure. In Fig. 3B, the authors showed that glucose uptake is unchanged in proximal tibia but decreased in distal tibia after 72-hour cold challenge. Considering the findings of the previous publication, it's also necessary to show whether the number/abundance of BMAT in different regions is changed after 72-hour cold, preferably using osmium staining. This will further explain whether the unchanged or decreased cold-induced glucose uptake in BMAT is due to altered BMAT abundance or BMAT is resistant to cold-induced glucose uptake indeed.

Thank you for highlighting this. The bone samples from our PET/CT animals were either frozen (for mRNA) or fixed and sectioned for histology, so it has not been possible to do osmium staining.

However, by histomorphometry we have now quantified the adipocyte density (N.Ad/Ma.Ar; number per mm² BM area) and adipose area (Ad.Ar/Ma.Ar, %) within the proximal and distal tibial BM; images of the proximal tibial BM are also now included in Figure 5D (previous figure 3D). The new histomorphometric data is presented in Figure S4 of the revised manuscript and shows that adipocyte density and BM adiposity in the proximal or distal tibia do not differ between the control, acute or chronic cold-exposed mice. Therefore, while Erica Scheller's previous work shows that 21-day cold exposure can markedly decrease tibial BMAT, our current studies show that 3 days of cold exposure is not sufficient to alter BM adiposity. We discuss this data in lines 249-255 and 380-390 of the revised manuscript.

2.5 Similar to Comment 4, in addition to glucose uptake, it's also necessary to show the CT images highlighting BMAT in humans at room temperature and cold environment in Figure 4, which will further consolidate your conclusion.

For these three groups of human subjects we have now created 3D renders to show the bone and BMAT in CT scans and have also compared measurements of bone marrow adiposity (Ad.V/Ma.V, %) in the humeri, femurs, clavicle, sternum and vertebrae of each group. These new data are presented in Figure 7A and S6B and described in lines 299-301 of the revised manuscript. They show that the groups do not significantly differ in Ad.V/Ma.V in the femur, clavicle, sternum or vertebrae, whereas in the humerus the cold-exposed group has slightly lower Ad.V/Ma.V than the room temperature (no BAT) group. It is possible that this decrease reflects BMAT catabolism in response to cold exposure, but we think this is unlikely given that this is a relatively mild cold exposure (2 h at 16 °C). Indeed, our mouse data shows that even 3 days at 4 °C is not sufficient to decrease BMAT (e.g. new Figure S4). As shown in our original submission, cold exposure also does not affect glucose uptake in BMAT of the humerus or at other skeletal sites (Figure 7B-D of revised manuscript). This supports our conclusion that BMAT in humans does not undergo cold-induced glucose uptake and is therefore distinct from BAT.

2.6 Could you explain which type of glucose transporter is responsible for basal glucose uptake in BMAT?

See also response to 1.4. We speculate that this is via SLC2A1 or SLC2A3. We address this in the Discussion (Lines 463-466) by stating, "*Why does BMAT have such high basal glucose uptake? This may result from BMATs having high expression of SLC2A1 and/or SLC2A3 (Fig. 1), a finding supported by previous microarray studies³⁵. Indeed, among numerous human tissues, SLC2A3 expression is highest in BM³³, while Slc2a3 is also greater in BM than in WAT of mice³⁴.*"

2.7 What is the contribution of BMAT-mediated basal glucose uptake to systemic glucose homeostasis?

To further address the contribution of BMAT to systemic glucose homeostasis we have followed the suggestion of Reviewer 3 (point 1A) by doing calculations to estimate the total glucose uptake capacity of BMAT, bone and red marrow in comparison to skeletal muscle (Figure R3). This analysis shows that BMAT has a total glucose uptake capacity that is up to 30% of that for skeletal muscle. This is greater than the uptake capacity of red marrow (5-10% of muscle uptake) but less than that of bone tissue (approx. 50-80% of muscle uptake). Further details are provided in our response to reviewer 3, and in lines 445-461 of the updated manuscript.

Other minor issues:

2.8 The resolution of PET/CT images (Fig. 2B, Fig. 2D, Fig. 3A) is too low. You should show images with higher resolution.

We agree that higher-resolution images would be preferable, but unfortunately this resolution is the highest output from the PET/CT software. Therefore, we are unable to show higher-resolution images. Instead we have updated the figures by showing mouse PET/CT images at slightly lower zoom to decrease the pixellation; for figure 5A (previous figure 3A) this was also done to better visualise the FDG uptake in the paraspinal region.

2.9 For Fig. 3D, you'd better use immunohistochemical staining of UCP1 rather than H&E staining. In addition, the image of iWAT after chronic cold exposure (Fig. 3D) didn't show obvious beige phenotype compared to acute cold and control groups. You need to show a more representative one.

Thank you for these suggestions. One of the co-authors of our paper, Dr Erica Scheller, has extensive experience with UCP1 immunostaining of BM, WAT and BAT. Her lab has recently published studies showing that UCP1 immunostaining is not reliable for BM sections owing to very high non-specific binding of this antibody (Craft et al., 2019). We also feel that the data in Figures 3E-G of our original submission (now Figures 5E-G) show very clearly that there is induction of *Ucp1* expression in BAT and iWAT but that *Ucp1* is not detectable in BMAT; based on this, it is highly unlikely that UCP1 protein would be detectable in BMAT by IHC.

For figure 3D we apologise if beigeing was not obvious in the iWAT micrographs. In our revised manuscript we have now included higher-magnification images of iWAT (Figure S4E), as well as quantification of the frequency of multilocular adipocytes in iWAT (Figure S4F). These updates demonstrate more clearly that there is a robust beigeing response in the iWAT, consistent with the PET/CT and molecular analyses shown in Figures 5C and 5F. Notably, no multilocular adipocytes were detectable in the BM, which further supports the conclusion that BMAT does not undergo a beigeing response.

Reviewer #3

In "Bone marrow adipose tissue is a unique adipose subtype with distinct roles in systemic glucose homeostasis" by Suchacki et al., the authors address fundamental questions related to the physiological relevance of bone marrow adipose tissue (BMAT). They first demonstrated in mouse tissue that BMAT has a different transcriptional profile compared to either white or brown and beige/brite tissues. Curiously, neither insulin nor cold exposure increased glucose uptake in BMAT, as assessed by both adipose tissue biopsies and 18F-FDG PET/CT, the latter being a novel approach to assessing an adipose tissue not easily accessible. They finally turned to human BMAT using 18F-FDG PET/CT and report that in the basal state, BMAT is a major site of glucose uptake.

The manuscript has several strengths, among which are the excellent execution of technically challenging experiments; the introduction of a method to study human BMAT in situ via non-invasive imaging; and the use of different animal and human models of BMAT to create a fuller picture of its physiological roles. It is striking that BMAT plays little role in glucose uptake in response to insulin, cold exposure, or even glucocorticoids, the last of which the authors described in previous publications.

The manuscript could be improved by addressing the following points:

3.1 Physiological significance of BMAT in rodents and humans. The authors appropriately support the part of the title stating that BMAT is a unique adipose tissue. What is not clear is how it could influence systemic glucose homeostasis. Much of the data provided actually suggest it is quite metabolically inert in that it does not respond to insulin, cold exposure, or glucocorticoids. It sounds as if the authors were unsatisfied with the extent of the advances they made in the understanding of BMAT physiology [nicely discussed in the first paragraph of the Discussion and other locations as well]. Instead, they concluded with a rather broad claim that BMAT has the “potential to influence systemic glucose homeostasis”.

Thank you for your thoughts here. The conclusion that BMAT might influence systemic glucose homeostasis is based on the unexpected observation that BMAT has high basal glucose uptake. This is particularly intriguing given that BMAT glucose uptake resists insulin stimulation and is not cold-responsive, a finding that is in no way dissatisfying because it shows, for the first time, that BMAT glucose metabolism *in vivo* is distinct from that in WAT or BAT. Our conclusions that this high basal glucose uptake might have systemic implications relate to the fact that BMAT glucose uptake is higher than uptake in bone (e.g. Fig. 7 of the revised manuscript), and that bone glucose uptake is required for normal metabolic function (Li et al., 2016). Moreover, we now present data from our updated analyses which reveal that SUVs for glucose uptake in BMAT are similar or greater than those for skeletal muscle; please see our reply to your next point for further details of this.

In addition, please see response to Reviewer 1.4 regarding why BMAd require such high rates of glucose uptake. We speculate that these are needed for *de novo* lipogenesis, with BMAd lipid stores then being required as a local energy source to support haematopoiesis. We now elaborate on this in lines 475-485 of the revised manuscript.

3.1A Quantification - Figs. 4F and 4G are critical to the paper vis-à-vis the human BMAT. The glucose uptake, as measured by SUV, is higher in sternum compared to bone and red marrow in 3 of the 5 sites studied; and glucose uptake in BMAT is higher than subcutaneous WAT. However, these are all very low SUV values. The authors report that BMAT can be >10% of total adipose tissue mass. They now need to show in grams of glucose how much glucose is taken up by the bone, RM, scWAT, and BMAT sites in the basal state. These values should be compared to skeletal muscle. It would be helpful if skeletal muscle SUV was shown in Fig. 2F. The authors can then discuss these absolute numbers and determine whether the BMAT depots represent a major site for glucose uptake in the basal state, both compared to other WAT depots as well as skeletal muscle. Either substantial or not, BMAT's glucose uptake amounts would be of importance.

We agree that this is an important issue in considering the potential contribution of BMAT to systemic glucose disposal. To determine how BMAT SUVs compare to those in skeletal muscle we have now analysed skeletal muscle SUVs for mice and humans. These data are presented in Figures 3, 5 and 7 of the revised manuscript. For Figure 3 (mouse insulin-treatment studies) this analysis confirms insulin-stimulated glucose uptake in the skeletal muscle but also shows that, in the vehicle-treated mice, muscle glucose uptake occurs at a similar rate to uptake in distal tibial BMAT and bone, and at a lower level than in bone or BM of the humerus, femur and proximal tibia. This latter finding is consistent with the results of Zoch *et al*, who find that rates of FDG uptake in mice are several fold higher in whole bones than in skeletal muscle (Zoch et al., 2016). For humans (Fig. 7F of the revised manuscript), SUVs in skeletal muscle are similar to those for BMAT in appendicular bones but lower than the SUVs in axial BMAT.

Absolute uptake capacity, in grams of glucose, is more difficult to measure reliably. The ideal way to do so would be through total-body PET, but unfortunately this technology is still in its infancy and currently is available in only a handful of institutions worldwide (Cherry et al., 2018). Therefore, to address this we have instead estimated the volumes of skeletal muscle, bone, BMAT and red marrow based on published reference ranges, taking into account the age, sex, body mass and height of each subject. These tissue volumes are shown in Figure R3.A of this response to the reviewers. Using these volumes, and the SUVs for each tissue, we then calculated the total glucose uptake for each tissue type. Here, an important consideration is that for bone, BMAT and RM, the SUVs for FDG uptake are greater in the axial skeleton than in the appendicular skeleton. Therefore, for these tissues we calculated two estimates for total glucose uptake: one based on SUVs from the axial skeleton and one based on SUVs from the appendicular skeleton.

These estimates of total glucose uptake in skeletal muscle, bone, BMAT and red marrow are presented in Figure R3.B and show that BMAT has a total glucose uptake capacity that is up to 30% of that for skeletal muscle. This is greater than the uptake capacity of red marrow (5-10% of muscle uptake) but slightly less than that of bone tissue (approx. 50-80% of muscle uptake). This demonstrates that, in addition to having high SUVs for glucose, BMAT also has a relatively high absolute capacity for glucose uptake that is likely to influence glucose homeostasis.

These estimates of absolute glucose uptake capacity are informative but, although they are based on published observations and our direct SUV measurements, they remain estimates. Indeed, without conducting total-body PET/CT it is not possible to accurately and reliably determine the capacity for absolute glucose uptake in these different tissues. Therefore, one concern is that presenting these data in the revised manuscript may cause readers to place undue confidence in the absolute values. For these reasons, we decided not to include these data in the revised manuscript, but we agree with you that the total glucose uptake capacity is a very important issue to address. Therefore, we have updated the Discussion of the manuscript (Lines 445-461) to elaborate on these issues, including many of the details provided above. We have also updated the Methods section (Lines 916-933) to describe how we estimated total tissue volumes and SUVs in humans. If the reviewers and editors feel that these calculations are essential to include in the Results then we would consider adding them as a supplemental figure. However, because they are estimates we think it is more appropriate to instead emphasise these issues in the Discussion.

3.1B How is lipogenesis in BMAT regulated if it is not insulin responsive?

This is another excellent question but the answer currently is not clear. There are ample data in the literature showing that BMAT, compared to WAT, has high fatty acid uptake and esterification and that this is unaltered during fasting or starvation. This suggests that, unlike in WAT, lipogenesis in BMAT is less sensitive to insulin regulation. One mechanism through which insulin stimulates lipogenesis is by suppressing AMPK activity. Thus, given that BMAT accumulates during caloric restriction, one possibility is that BMAT may have lower AMPK activity than WAT and that this results in BMAT having a higher rate of basal lipogenesis that is less dependent on insulin regulation. In addition, we show in revised Figure 4 that insulin can stimulate Akt-S473 phosphorylation in BMAT (albeit less than in WAT), which is notable because Akt phosphorylation at S473 is necessary for lipogenesis (Hagiwara et al., 2012). In separate studies we have also identified several transcriptional differences that might explain the insulin-independent regulation of BMAT lipogenesis. These and other possibilities go beyond the scope of the current manuscript and are something that we are beginning to address using PET/CT and other approaches. In the

revised manuscript we have updated the Discussion (Lines 355-365 and 475-485) to more clearly highlight these questions about insulin and BMAT lipogenesis.

3.1C What could the physiological roles of BMAT be if it does not respond to critical hormonal regulators such as insulin?

Our conclusion is not that BMAT is completely unresponsive to insulin, but only that it is less responsive than WAT to insulin-stimulated glucose uptake. This is analogous to BMAT's response to catecholamine-stimulated lipolysis: BMAT can undergo such lipolysis, but resists this in comparison to WAT (Scheller et al., 2018). We speculate that BMAT is acting as a local energy reserve for haematopoiesis and that, therefore, this function needs to be consistent even in the face of fluctuations in food intake and insulin. This is reminiscent of the perinodal adipocytes that surround lymph nodes, whose resistance to systemic hormonal cues allows these adipocytes to maintain consistent local interactions with lymphoid tissue (Craft and Scheller, 2017; Mattacks and Pond, 1999). Thus, BMATs' relative insulin resistance may allow their haematopoietic-supporting function to be unperturbed in the face of altered food intake and insulin levels. However, because BMAT has such high basal uptake of glucose and fatty acids, this local metabolic function has the potential to influence systemic glucose and lipid metabolism. We elaborate on this in lines 463-473 of the Discussion.

3.2 Clinical trial information

3.2A Was this clinical trial registered with any central database such as the EU Clinical Trials Register?

We have updated the clinical trial information in the Methods (lines 655-658). The updated reporting summary now also includes all the necessary ethics and clinical trial information, as follows:

A. Subjects undergoing paired MRI and CT scans (Fig. 6A-C of revised manuscript) were from the following trials:

(1) The SALTIRE 2 trial (Bisphosphonates and RANKL Inhibition in Aortic Stenosis, NCT02132026) recruited patients aged over 50 years with a peak aortic jet velocity of >2.5 m/s and grade 2-4 calcification of the aortic valve on echocardiography.

(2) The PRE18FFIR trial (Prediction of recurrent events with ¹⁸F-fluoride to identify ruptured and high-risk coronary artery plaques in patients with myocardial infarction, NCT02278211) recruited patients with recent myocardial infarction and multi-vessel coronary artery disease on invasive angiography. Exclusion criteria for both trials include inability to receive iodinated contrast, renal impairment (estimated glomerular filtration rate ≤ 30 mL/min/1.73 m²) or women of child-bearing potential.

Characteristics of these subjects are provided in Table 1 of the Methods section.

B. Cold-exposed subjects (Fig. 7A-F, S6A-B of revised manuscript):

Healthy volunteers were exposed to a mild cold (16-7°C) for 2 h prior to their ¹⁸F-FDG PET/CT scan, as described previously (Weir et al., 2018). The study was approved by the South East Scotland Research Ethics Committee (ethics number 13/SS/0242).

C. Subjects with or without detectable BAT at room temperature (Fig. 6D-E, 7A-F and S6A-B of revised manuscript).

Characteristics are provided in Table 1 of the Methods section.

D. Subjects with placebo or prednisolone treatment (Fig. S6C of revised manuscript):

To assess effects of prednisolone treatment, six healthy young men (aged 22.1 +/- 1.2 years; BMI of 22.0 +/- 0.9) were recruited to a double-blind, randomised crossover study, as described previously (Ramage et al., 2016). The study was approved by the South East Scotland Research Ethics Committee (ethics number 11/SS/0074).

3.2B Was the number of human subjects studied prespecified? Was a sample size calculation or at least a detectable difference calculation done?

The cold and GC-treated patients were based on power calculations to determine sample sizes. 'Active BAT' patients at room temperature were identified retrospectively, and their 'no BAT' counterparts were then identified to match subject characteristics with the 'Active BAT' group. No sample size calculations were done because these analyses depended on retrospective identification of the 'Active BAT' group. We have now updated the Methods (lines 660-663) to clarify these points.

Minor comments:

3.3 In graphs where mRNA expression is lower than 1.0, it would be helpful to use a log scale. Examples are 1G, 3E-G, and S3F-I.

We tried changing these graphs to a log scale however we felt the way we presented the graphs initially was easier to interpret. For these we purposefully analysed the qPCR data so that, for each transcript, the group with the highest expression has an expression value of 1.0. Therefore, necessarily all other groups (for that transcript) will have an expression of lower than 1.0. This method allows different transcripts, with different expression patterns, to be easily visualised on the same scale of y-axis.

3.4 it is not clear what the highest value is before the axis break. Is it 2.0?

Sorry for any confusion. We have now updated the axis break of Figure 5B (previous Figure 3B). You are correct that in the original manuscript the lower axis segment went up to 2.0, but in the revised manuscript we have this going up to 3.0 so that the basal BAT uptake can be more easily compared to basal uptake in the other tissues. Thank you for highlighting this.

3.5 The phrase should be "distinct from" not "distinct to".

Thanks, we have now changed the text where applicable to "distinct from"

References cited in this response to reviewers:

- Attané, C., Estève, D., Chaoui, K., Iacovoni, J., Corre, J., Moutahir, M., Valet, P., Schiltz, O., Reina, N., and Muller, C. (2019). Yellow adipocytes comprise a new adipocyte sub-type present in human bone marrow. *bioRxiv*, 641886.
- Bartelt, A., Koehne, T., Todter, K., Reimer, R., Muller, B., Behler-Janbeck, F., Heeren, J., Scheja, L., and Niemeier, A. (2017). Quantification of Bone Fatty Acid Metabolism and Its Regulation by Adipocyte Lipoprotein Lipase. *Int J Mol Sci* *18*.
- Bathija, A., Davis, S., and Trubowitz, S. (1979). Bone marrow adipose tissue: response to acute starvation. *Am J Hematol* *6*, 191-198.
- Cawthorn, W.P., Scheller, E.L., Parlee, S.D., Pham, H.A., Learman, B.S., Redshaw, C.M., Sulston, R.J., Burr, A.A., Das, A.K., Simon, B.R., *et al.* (2016). Expansion of Bone Marrow Adipose Tissue During Caloric Restriction Is Associated With Increased Circulating Glucocorticoids and Not With Hypoleptinemia. *Endocrinology* *157*, 508-521.
- Cherry, S.R., Jones, T., Karp, J.S., Qi, J., Moses, W.W., and Badawi, R.D. (2018). Total-Body PET: Maximizing Sensitivity to Create New Opportunities for Clinical Research and Patient Care. *Journal of nuclear medicine : official publication, Society of Nuclear Medicine* *59*, 3-12.
- Craft, C.S., Li, Z., MacDougald, O.A., and Scheller, E.L. (2018). Molecular differences between subtypes of bone marrow adipocytes. *Curr Mol Biol Rep* *4*, 16-23.
- Craft, C.S., Robles, H., Lorenz, M.R., Hilker, E.D., Magee, K.L., Andersen, T.L., Cawthorn, W.P., MacDougald, O.A., Harris, C.A., and Scheller, E.L. (2019). Bone marrow adipose tissue does not express UCP1 during development or adrenergic-induced remodeling. *Scientific reports* *9*, 17427.
- Craft, C.S., and Scheller, E.L. (2017). Evolution of the Marrow Adipose Tissue Microenvironment. *Calcif Tissue Int* *100*, 461-475.
- Cypess, A.M., Lehman, S., Williams, G., Tal, I., Rodman, D., Goldfine, A.B., Kuo, F.C., Palmer, E.L., Tseng, Y.H., Doria, A., *et al.* (2009). Identification and importance of brown adipose tissue in adult humans. *N Engl J Med* *360*, 1509-1517.
- Dezso, Z., Nikolsky, Y., Sviridov, E., Shi, W., Serebriyskaya, T., Dosymbekov, D., Bugrim, A., Rakhmatulin, E., Brennan, R.J., Guryanov, A., *et al.* (2008). A comprehensive functional analysis of tissue specificity of human gene expression. *BMC biology* *6*, 49.
- Engfeldt, P., Arner, P., and Ostman, J. (1980). Influence of adipocyte isolation by collagenase on phosphodiesterase activity and lipolysis in man. *J Lipid Res* *21*, 443-448.
- Gesta, S., Lolmede, K., Daviaud, D., Berlan, M., Bouloumie, A., Lafontan, M., Valet, P., and Saulnier-Blache, J.S. (2003). Culture of human adipose tissue explants leads to profound alteration of adipocyte gene expression. *Horm Metab Res* *35*, 158-163.
- Hagiwara, A., Cornu, M., Cybulski, N., Polak, P., Betz, C., Trapani, F., Terracciano, L., Heim, Markus H., Rüegg, Markus A., and Hall, Michael N. (2012). Hepatic mTORC2 Activates Glycolysis and Lipogenesis through Akt, Glucokinase, and SREBP1c. *Cell Metab* *15*, 725-738.
- Kozubik, A., Sedlakova, A., Pospisil, M., and Petrasek, R. (1988). In vivo studies of the relationship between the activation of lipid metabolism, postirradiation bone marrow cell proliferation and radioresistance of mice. *Gen Physiol Biophys* *7*, 293-302.
- Li, Z., Frey, J.L., Wong, G.W., Faugere, M.C., Wolfgang, M.J., Kim, J.K., Riddle, R.C., and Clemens, T.L. (2016). Glucose Transporter-4 Facilitates Insulin-Stimulated Glucose Uptake in Osteoblasts. *Endocrinology* *157*, 4094-4103.
- Liu, L.F., Shen, W.J., Ueno, M., Patel, S., and Kraemer, F.B. (2011). Characterization of age-related gene expression profiling in bone marrow and epididymal adipocytes. *BMC Genomics* *12*, 212.
- Mattacks, C.A., and Pond, C.M. (1999). Interactions of noradrenalin and tumour necrosis factor alpha, interleukin 4 and interleukin 6 in the control of lipolysis from adipocytes around lymph nodes. *Cytokine* *11*, 334-346.
- Mattiucci, D., Maurizi, G., Izzi, V., Cenci, L., Ciarlantini, M., Mancini, S., Mensa, E., Pascarella, R., Vivarelli, M., Olivieri, A., *et al.* (2018). Bone marrow adipocytes support hematopoietic stem cell survival. *J Cell Physiol* *233*, 1500-1511.
- Pazzaglia, U.E., Congiu, T., Raspanti, M., Ranchetti, F., and Quacci, D. (2009). Anatomy of the intracortical canal system: scanning electron microscopy study in rabbit femur. *Clin Orthop Relat Res* *467*, 2446-2456.
- Ramage, L.E., Akyol, M., Fletcher, A.M., Forsythe, J., Nixon, M., Carter, R.N., van Beek, E.J., Morton, N.M., Walker, B.R., and Stimson, R.H. (2016). Glucocorticoids Acutely Increase Brown Adipose Tissue Activity in Humans, Revealing Species-Specific Differences in UCP-1 Regulation. *Cell Metab* *24*, 130-141.

- Rao, R.R., Long, J.Z., White, J.P., Svensson, K.J., Lou, J., Lokurkar, I., Jedrychowski, M.P., Ruas, J.L., Wrann, C.D., Lo, J.C., *et al.* (2014). Meteorin-like is a hormone that regulates immune-adipose interactions to increase beige fat thermogenesis. *Cell* *157*, 1279-1291.
- Rosell, M., Kaforou, M., Frontini, A., Okolo, A., Chan, Y.W., Nikolopoulou, E., Millership, S., Fenech, M.E., MacIntyre, D., Turner, J.O., *et al.* (2014). Brown and white adipose tissues: intrinsic differences in gene expression and response to cold exposure in mice. *Am J Physiol Endocrinol Metab* *306*, E945-964.
- Scheller, E.L., Khandaker, S., Learman, B.S., Cawthorn, W.P., Anderson, L.M., Pham, H.A., Robles, H., Wang, Z., Li, Z., Parlee, S.D., *et al.* (2018). Bone marrow adipocytes resist lipolysis and remodeling in response to beta-adrenergic stimulation. *Bone* *118*, 32-41.
- Shafat, M.S., Oellerich, T., Mohr, S., Robinson, S.D., Edwards, D.R., Marlein, C.R., Pidcock, R.E., Fenech, M., Zaitseva, L., Abdul-Aziz, A., *et al.* (2017). Leukemic blasts program bone marrow adipocytes to generate a protumoral microenvironment. *Blood* *129*, 1320-1332.
- Shin, H., Ma, Y., Chanturiya, T., Cao, Q., Wang, Y., Kadegowda, A.K.G., Jackson, R., Rumore, D., Xue, B., Shi, H., *et al.* (2017). Lipolysis in Brown Adipocytes Is Not Essential for Cold-Induced Thermogenesis in Mice. *Cell Metab* *26*, 764-777.e765.
- Shore, A.M., Karamitri, A., Kemp, P., Speakman, J.R., Graham, N.S., and Lomax, M.A. (2013). Cold-induced changes in gene expression in brown adipose tissue, white adipose tissue and liver. *PLoS One* *8*, e68933.
- Tavassoli, M. (1976). Marrow adipose cells. Histochemical identification of labile and stable components. *Arch Pathol Lab Med* *100*, 16-18.
- Thorrez, L., Van Deun, K., Tranchevent, L.C., Van Lommel, L., Engelen, K., Marchal, K., Moreau, Y., Van Mechelen, I., and Schuit, F. (2008). Using ribosomal protein genes as reference: a tale of caution. *PLoS One* *3*, e1854.
- Trubowitz, S., and Bathija, A. (1977). Cell size and plamitate-1-14c turnover of rabbit marrow fat. *Blood* *49*, 599-605.
- Weir, G., Ramage, L.E., Akyol, M., Rhodes, J.K., Kyle, C.J., Fletcher, A.M., Craven, T.H., Wakelin, S.J., Drake, A.J., Gregoriades, M.L., *et al.* (2018). Substantial Metabolic Activity of Human Brown Adipose Tissue during Warm Conditions and Cold-Induced Lipolysis of Local Triglycerides. *Cell Metab* *27*, 1348-1355 e1344.
- Yusuf, R.Z., and Scadden, D.T. (2012). Fate through fat: lipid metabolism determines stem cell division outcome. *Cell Metab* *16*, 411-413.
- Zakaria, E., and Shafrir, E. (1967). Yellow bone marrow as adipose tissue. *Proc Soc Exp Biol Med* *124*, 1265-1268.
- Zoch, M.L., Abou, D.S., Clemens, T.L., Thorek, D.L., and Riddle, R.C. (2016). In vivo radiometric analysis of glucose uptake and distribution in mouse bone. *Bone Res* *4*, 16004.

Figure R1

Figure R1 – adipocyte isolation from scWAT and BM of humans. Adipocytes isolated from scWAT or BM (human cohort 2) were stained with Hoescht and LipidTox DeepRed (ThermoFisher, cat. no. H34477). Stained cells were then analysed using a Nexcelom Vision Cellometer. Images confirm that all nuclei are associated with unilocular lipid droplets and that no other nucleated cells are present. Trabecular bone samples gave lower yields of adipocytes and therefore all of these samples were used for RNA isolation, rather than Nexcelom analysis, to ensure that sufficient RNA could be obtained for downstream analysis.

Figure R2

Figure R2– Effects of insulin on glucose uptake in explants of tibial BMAT and gWAT from rabbits. Glucose uptake in explants of marrow or adipose tissue. Distal tibial BMAT and gWAT were isolated from male New Zealand White rabbits and stored in Hanks Balanced Salt Solution (HBSS) at 37 °C. Tissues were then divided into 7 explants of approx. 40 mg each and were washed in warm HBSS. Explants were then transferred to separate wells of a 48-well plate and were serum-starved in warm Krebs Ringer Hepes (KRH) buffer + BSA (0.1%; fatty acid-free) for two hours. Explants were then washed and 500 μ L fresh KRH + BSA was added to each well. Explants were then treated with 100 nM insulin or with vehicle control (PBS). Ten minutes later, cytochalasin B was added to one well to block glucose uptake (providing a background control). After another 10 minutes, 14 C-deoxyglucose was added to each well. Glucose uptake was terminated 20 minutes later, i.e. 40 minutes post-insulin or vehicle treatment, by putting plates on ice and washing thrice with ice-cold PBS. Explants were then lysed in 0.3% SDS and glucose uptake analyzed by liquid scintillation counting. Counts were then normalized to the amount of protein in each explant sample. Four rabbits were used. Statistically significant differences were assessed by 2-way ANOVA with Sidak post-test for multiple comparisons. Within each tissue there were no significant differences between vehicle and insulin-treated explants. Within the vehicle-treated samples BMAT showed a trend for greater glucose uptake than gWAT. Samples from the same rabbit are indicated by the pattern of the lines between the vehicle and insulin points.

Reviewers' Comments:

Reviewer #1:

Remarks to the Author:

Very nice responses to my concerns. In Fig R1 I would like to see a scale bar. Also I still wonder why (if?) you cannot use another imaging microscope instead of the Nexcelom, which you explain is very low resolution. Also some other controls for your imaging would be good- aka lipid droplets alone (no cells) or other types of cells with no lipid droplets to validate the LipidTox stain. I have seen a lot of autofluorescence and I recommend you to be sure of what you are seeing. Why did you not use a confocal microscope? These are the best images I have seen of pBMAs, but I still wonder why it only seems to work with this microscope. This is not essential to me for the publication of your paper.

-Michaela Reagan

Reviewer #2:

Remarks to the Author:

The revised version has adequately addressed most concerns raised in my previous comments. However, there are a couple of minor issues which need to be addressed further.

Minor points:

2.2.1 In Figure 4A, the western blot image of 6th sample in gWAT group is not very satisfactory. There is only half lane for this sample. A better image is needed to replace the current one. In addition, since the basal level of Akt P-S473 in dBMAT is much lower than that in gWAT, and it seems that the fold change of Akt phosphorylation at S473 after insulin stimulation in dBMAT (~ 7 fold) is even higher than that in gWAT (~ 3 fold), this data is not a strong evidence which can conclude that dBMAT is markedly resistant to insulin-stimulated phosphorylation of Akt at S473.

2.3 Acute cold challenge should increase Pnpla2 (encoding adipose triglyceride lipase) and Lipe (encoding hormone-sensitive lipase) in iWAT, but this trend is not observed in your data presented in Figure S5B.

Reviewer #3:

Remarks to the Author:

The authors have done an excellent job in responding to the original set of comments.

NCOMMS-19-23835A – Second round of reviewers' comments

Reviewer #1

Very nice responses to my concerns. In Fig R1 I would like to see a scale bar. Also I still wonder why (if?) you cannot use another imaging microscope instead of the Nexcelom, which you explain is very low resolution. Also some other controls for your imaging would be good- aka lipid droplets alone (no cells) or other types of cells with no lipid droplets to validate the LipidTox stain. I have seen a lot of autofluorescence and I recommend you to be sure of what you are seeing. Why did you not use a confocal microscope? These are the best images I have seen of pBMAs, but I still wonder why it only seems to work with this microscope. This is not essential to me for the publication of your paper.

Thank you for these comments and suggestions. We have now updated the images in Figure R1 to include a scale bar. We have also added brightfield and fluorescent images of stromal-vascular cells stained with Hoescht and LipidTox. These show that there is no red fluorescence signal in the absence of lipid-laden cells, validating the specificity of the LipidTox stain.

Our reason for using the Nexcelom Cellometer was that we wanted a system that could automate the counting and analysis of our isolated cells. In particular, we were interested in assessing the percentage of nuclei in the adipocyte fractions that were not associated with lipid droplets, and also automated analysis of adipocyte size distribution in the samples from WAT and BAT. The former was of interest to assess the purity of the isolated adipocytes, while the latter was of interest to test if we saw differences in the sizes of the white and BM adipocytes (e.g. as reported in previous publications). Unfortunately, we found that the Nexcelom software was very poor at detecting adipocytes, often including only a small % of cells apparent in each image. However, by this stage it was clear that we consistently obtained sufficiently pure adipocyte preparations, so we never pursued confocal as an alternative approach for this. However, in future we are likely to use confocal so that we can obtain higher-resolution images.

Reviewer #2

The revised version has adequately addressed most concerns raised in my previous comments. However, there are a couple of minor issues which need to be addressed further.

Minor points:

2.2.1 In Figure 4A, the western blot image of 6th sample in gWAT group is not very satisfactory. There is only half lane for this sample. A better image is needed to replace the current one. In addition, since the basal level of Akt P-S473 in dBMAT is much lower than that in gWAT, and it seems that the fold change of Akt phosphorylation at S473 after insulin stimulation in dBMAT (~7 fold) is even higher than that in gWAT (~3 fold), this data is not a strong evidence which can conclude that dBMAT is markedly resistant to insulin-stimulated phosphorylation of Akt at S473.

We agree that a better image would be preferable. Unfortunately, we no longer have any remaining protein lysates because most of these were used in experiments for a previous publication (Scheller et al., 2018)). Therefore, we are unable to repeat these immunoblots. However, it is important to note that the loading for gWAT sample 6 has not compromised the quantification of Akt phosphorylation or expression, because this issue also applies to the loading controls (Akt, ERK1/2 and alpha-Tubulin). This is apparent from the raw data (shown below), which are included in the Source Data file. These raw data confirm that the gWAT values for Akt phosphorylation in Rat 6 are within the range of the other vehicle-treated gWAT samples.

		Akt P-S473							
		Vehicle				Insulin			
		Rat 1	Rat 2	Rat 5	Rat 6	Rat 3	Rat 4	Rat 7	Rat 8
gWAT		2.339	0.929	0.273	0.458	6.093	3.071	2.434	1.771
BMAT		0.117	0.131	0.090	0.095	0.849	0.979	0.312	0.432

		Akt P-T308							
		Vehicle				Insulin			
		Rat 1	Rat 2	Rat 5	Rat 6	Rat 3	Rat 4	Rat 7	Rat 8
gWAT		1.759	0.197	0.676	1.368	0.830	5.069	5.349	4.673
BMAT		0.011	0.049	0.383	0.083	0.213	0.046	0.090	0.097

		Akt (total)							
		Vehicle				Insulin			
		Rat 1	Rat 2	Rat 5	Rat 6	Rat 3	Rat 4	Rat 7	Rat 8
gWAT		0.465	0.681	1.300	1.554	0.721	0.862	1.969	1.843
BMAT		0.966	0.932	1.535	1.692	0.742	0.772	3.065	1.369

Table showing raw values from densitometry, after normalizing to loading controls.

Regarding Akt S473 phosphorylation in BMAT, you raise an important point about the relative fold increase between insulin-treated and vehicle-treated samples. From this, we agree that it is inappropriate to conclude that BMAT is resistant to this effect of insulin, and therefore we have updated the text of the revised manuscript as follows:

Abstract, Line 54: "BMAT resists insulin-stimulated Akt T308 phosphorylation"

Results, Lines 230-236: "*In both control and insulin-treated rats, the absolute levels of Akt S473 and T308 phosphorylation were lower in dBMAT than in gWAT; however, in dBMAT stimulation of S473 phosphorylation was readily apparent (Fig. 4A-B). In contrast, in dBMAT insulin did not affect T308 phosphorylation, despite similar total Akt expression between these two tissues (Fig. 4A-B). These data confirm and extend our PET/CT results by demonstrating that BMAT resists insulin-stimulated phosphorylation of Akt at T308, a key step of the insulin signalling pathway.*"

Discussion, Lines 363-364: "*We show that BMAT is capable of insulin-stimulated Akt S473 phosphorylation*".

Legend, Figure 4: Updated title to "BMAT resists insulin-stimulated Akt T308 phosphorylation"

2.3 Acute cold challenge should increase Pnpla2 (encoding adipose triglyceride lipase) and Lipe (encoding hormone-sensitive lipase) in iWAT, but this trend is not observed in your data presented in Figure S5B.

We also expected acute cold to increase expression of these transcripts in iWAT and therefore were initially surprised by the lack of increase. However, our finding is consistent with several other papers that also find no induction of these transcripts in response to cold exposure (Hao et al., 2015; Shore et al., 2013). It is unclear why this effect of cold should vary between different studies, but it suggests that other transcripts within iWAT are more robust targets for the effects of cold exposure.

To clarify this issue, we have updated lines 259-261 of the revised manuscript by stating, “Surprisingly, cold exposure did not upregulate the lipolysis-related transcripts *Pnpla2* or *Lipe* in BAT or iWAT (Fig. S5A-B); however, this is consistent with several previous studies^{30,31}.”

Reviewer #3

The authors have done an excellent job in responding to the original set of comments.

Thank you, we appreciate your useful suggestions and your time in reviewing our manuscript.

References cited in this response to reviewers:

- Hao, Q., Yadav, R., Basse, A.L., Petersen, S., Sonne, S.B., Rasmussen, S., Zhu, Q., Lu, Z., Wang, J., Audouze, K., *et al.* (2015). Transcriptome profiling of brown adipose tissue during cold exposure reveals extensive regulation of glucose metabolism. *Am J Physiol Endocrinol Metab* 308, E380-392.
- Scheller, E.L., Khandaker, S., Learman, B.S., Cawthorn, W.P., Anderson, L.M., Pham, H.A., Robles, H., Wang, Z., Li, Z., Parlee, S.D., *et al.* (2018). Bone marrow adipocytes resist lipolysis and remodeling in response to beta-adrenergic stimulation. *Bone* 118, 32-41.
- Shore, A.M., Karamitri, A., Kemp, P., Speakman, J.R., Graham, N.S., and Lomax, M.A. (2013). Cold-induced changes in gene expression in brown adipose tissue, white adipose tissue and liver. *PLoS One* 8, e68933.

Figure R1.2 (version 2)

Figure R1.1 – adipocyte and SVC isolation from scWAT and BM of humans. Adipocytes isolated from scWAT or BM (human cohort 2) were stained with Hoescht and LipidTox DeepRed (ThermoFisher, cat. no. H34477). Stromal-vascular cells (SVCs) were also included as a negative control for LipidTox staining. Stained cells were then analysed using a Nexcelom Vision Cellometer. Images confirm that, in the adipocyte samples, all nuclei are associated with unilocular lipid droplets and that no other nucleated cells are present. Analysis of the stromal vascular cells confirms that the LipidTox signal is dependent on the presence of lipid droplets. Trabecular bone samples gave lower yields of adipocytes and therefore all of these samples were used for RNA isolation, rather than Nexcelom analysis, to ensure that sufficient RNA could be obtained for downstream analysis.

Reviewers' Comments:

Reviewer #2:

Remarks to the Author:

The revised manuscript has addressed my concerns. thank you